# Fractionally charged particles at the energy frontier: The SM gauge group and one-form global symmetry

**Seth Koren⋆ and Adam Martin†**

Department of Physics and Astronomy, University of Notre Dame,
South Bend, IN, 46556 USA

⋆ skoren@nd.edu , † amarti41@nd.edu

## Abstract

The observed Standard Model is consistent with the existence of vector-like species with electric charge a multiple of $e/6$. The discovery of a fractionally charged particle would provide nonperturbative information about Standard Model physics, and furthermore rule out some or all of the minimal theories of unification. We discuss the phenomenology of such particles and focus particularly on current LHC constraints, for which we reinterpret various searches to bound a variety of fractionally charged representations. We emphasize that in some circumstances the collider bounds are surprisingly low or nonexistent, which highlights the discovery potential for these species which have distinctive signatures and important implications. We additionally offer pedagogical discussions of the representation theory of gauge groups with different global structures, and separately of the modern framework of Generalized Global Symmetries, either of which serves to underscore the bottom-up importance of these searches.

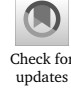

# 1  Introduction

**The fundamental charge quantum of QED**   What is the fundamental quantum of electric charge in the infrared quantum electrodynamics of our universe? This is an important particle physics question which is as yet unresolved. The Bayesian prior of high energy theory orthodoxy expects the answer to be $e$, the electric charge of the electron. If the Standard Model fields are *ever* unified in $SU(5)$ or $SO(10)$, this is necessarily true.[1]

But a lesson one could contemplate from recent decades of Beyond the Standard Model physics is that grand theories about the ultraviolet which we have come to love seem not to be realized in quite the way we thought. We have not produced superparticles, nor directly detected dark matter, nor found exotic kaon decays, nor observed an electron electric dipole moment. And we have not seen protons decay. We should indeed always be questioning which of our cherished principles to cling to, and which to consider counterfactually.

Notably, with less ambitious unification schemes we can have a smaller quantum of electric charge. As examples, in Pati-Salam theories (where we do not have full gauge coupling unification) the fundamental infrared charge can be $e/2$, and in theories of trinification (where we must add additional fermions) the quantum can be $e/3$. If the Standard Model matter never organizes into one of these minimal unified theories, then the fundamental quantum of charge can be $e/6$.[2] In more exotic scenarios that would even more generally challenge our usual UV paradigms, the charge could be even smaller.

The core message of our work is that particles with $\mathcal{O}(1)$ electric charges are an important probe of ultraviolet physics which have a universal infrared understanding. And it is not unreasonable to believe that they could exist near the electroweak scale to be found at the energy frontier. After all, we have only recently uncovered the full chiral spectrum of the Standard Model; it is certainly possible that this matter content cannot tell apart different UV scenarios but that our discovery of the least-massive vector-like states will distinguish them further.

---

[1]For a reminder of the experimental and theoretical reasons which would point one toward this preference, see Witten's beautiful 2002 Heinrich Hertz Lecture 'Quest for Unification' [1].

[2]Early work on extended models of unification which feature fractionally charged particles includes [2–9], and early discussions of the appearance of fractionally charged particles in string theories include [10–13].

Table 1: Colored particles in the Standard Model after electroweak symmetry breaking. $i$ is the generation index and here we use Dirac fermions. The charge is given in units of $e$.

|  | $u_i$ | $d_i$ | $g$ |
|---|---|---|---|
| $SU(3)_C$ | **3** | **3** | **8** |
| $U(1)_{\text{EM}}$ | $\frac{2}{3}$ | $-\frac{1}{3}$ | $0$ |

One may be misled into thinking that the question of the smallest charge of quantum electrodynamics is ultimately a question about *normalization*, and should not make much difference physically. It is true that the perturbative physics of QED is not modified in any case. But the nonperturbative physics *is* modified, as we will discuss in detail below.

And while the nonperturbative physics of the Standard Model is difficult to access with only the SM degrees of freedom, the discovery of a new particle can reveal nonperturbative aspects of the Standard Model physics. We learn that the allowed charges of magnetic monopoles, the spectrum of fractional instantons, and the possible Aharonov-Bohm phases are all modified. And as we have just said, the possibilities for the Standard Model species to unify in the ultraviolet depend crucially on this nonperturbative physics. This means that determining the fundamental charge quantum of QED could falsify large classes of models of grand unification, or potentially all of them.

**From QCD to QED**   Do not be confused by the charges of the quarks—by quantum electrodynamics we mean a long-distance theory far below the scale of confinement where the degrees of freedom are leptons and hadrons. The particular pattern of Yang-Mills representations we see borne out in the Standard Model unavoidably implies that all colorless hadrons have charge quantized in units of $e$, the electron's charge.

We can see this with a quick representation-theoretic argument, and we'll understand what's happening more generally in Section 6. Let us begin with the Standard Model having flowed to energies below electroweak symmetry breaking. At these energies it is sensible to speak of quarks as Dirac fermions, as in Table 1. Of the known colored particles, each quark $\psi_i^a$ in the fundamental **3** representation has electric charge $q_i$ which obeys $3q_i = 2 \pmod 3$, and their antiparticles the $\bar{\mathbf{3}}$ anti-fundamental $\bar{\psi}_{jb}$ necessarily have $3q_j = 1 \pmod 3$. The gluons in the adjoint **8** are of course electrically uncharged.

The only invariant tensors of $SU(3)_C$ are $\delta_b^a$, $\varepsilon^{abc}$, and $\varepsilon_{abc}$, and we seek to build composite operators which are colorless. Working $\pmod 3$, we see $\delta_b^a$ pairs a 1 with a 2, and the Levi-Civita symbol composes three of the same charge—either way resulting in an electric charge $\sum 3q_i = 0 \pmod 3$. Dividing through by three, this is exactly the condition that every hadron has electric charge an integer multiple of $e$. For an arbitrarily complex bound state, ultimately color indices can only be contracted in these ways, and the same argument applies.

So with the particles of the Standard Model, there are *no* asymptotic states with fractional charge. But it is not clear from this argument whether this fundamentally must be the case, or whether this relationship might be broken once we discover new BSM particles. Indeed we do not know the answer, which ultimately must be settled by empirical data. We can understand the issue systematically and gauge-invariantly as being a question about a certain generalized symmetry which infrared physics may or may not have.

**Generalized Global Symmetries**   While the local, perturbative physics is not modified by the charge quantum, the nonperturbative physics certainly is. A useful strategy to understand these aspects systematically is by enlarging our notion of symmetries to include symmetries of

extended operators that appear in our field theories, such as Wilson and 't Hooft loops. Symmetries that act on such one-dimensional line operators are known as 'one-form symmetries'—to be contrasted with symmetries that act on local, point operators which are called 'zero-form symmetries'.

From the modern field theory perspective, which such one-dimensional gauge-invariant operators exist is part of the data needed to define a quantum field theory [14–19]. As a basic picture one can think of these operators as accessing the response of the system to a probe particle in a particular representation in the limit where the probe particle is infinitely massive so that it has a well-defined worldline. Note that we do not specify that the worldline must be a geodesic, or even timelike. With a spacelike worldline, one is familiar with using a Wilson loop operator $\exp(i \oint_\gamma A)$ to understand the Aharonov-Bohm effect where we think about adiabatically moving an electron on the spatial path $\gamma$ around a solenoid (or possibly a cosmic string).

As such, to fully understand the quantum field theory describing the particles of the Standard Model, we must also analyze the symmetries of the one-dimensional gauge-invariant operators we can write down, whether in the electroweak phase or at lower energies. In the full Standard Model the different 'global structures of the gauge group' (to be reviewed below) are exactly the question of whether the Standard Model has a discrete group of electric one-form global symmetries, or whether (some of) these electric one-form symmetries should actually be gauged to instead produce extra *magnetic* one-form global symmetries. This trade-off is as could be expected from Dirac quantization.

Furthermore, this generalized symmetry language will provide a unifying, general understanding of what we learn from experimentally probing the existence of fractionally charged particles at the energy frontier. The question of the charge quantum of quantum electrodynamics can be rephrased universally in terms of emergent global electric one-form symmetry. We will introduce these concepts pedagogically in Section 7.

Such one-form symmetries are data about the field theory which are in some sense nonperturbative. That is, they are needed to have a more refined understanding of the Yang-Mills theory which goes past what minimal coupling, a Lagrangian procedure which only knows about local fields, depends upon. The Lagrangian depends only on perturbative data which are local in field space. In order to learn information about the global structure of the field space, we must have data which allow us to probe *paths* in field space, not just points. This is why there is new understanding to be gained by thinking about extended operators in our QFTs.[3]

**The energy frontier** As we have motivated above, searches for fractionally charged particles are some of the highest stakes experimental probes we have at the energy frontier. The observation of a particle with electric charge $e/6$, be it fundamental or hadronic, fermionic or bosonic, would unequivocally falsify all minimal grand unified theories. Perhaps no other single new particle discovery could teach us so much about the far ultraviolet of our universe, so it is well worth devoting experimental effort to searching for such particles.

Great energy frontier searches sensitive to fractionally charged particles have been undertaken in recent years by CMS (e.g. [20]) and ATLAS (e.g. [21]) but efforts have mainly been focused on SUSY-motivated scenarios. To the extent that we can design searches sensitive to the electric charges, fractionally charged particles can provide extremely distinctive signatures, since as discussed above there are strictly no particles with these properties in the Standard Model. We take here a first step toward a more general paradigm by reinterpreting existing

---

[3]Of course it is also natural to think about maps of *higher*-dimensional manifolds into field space, and one may indeed talk about $n$-dimensional operators and $n$-form symmetries, but in this work we will only use the concepts of Wilson and 't Hooft lines and their 1-form symmetries.

searches for various benchmark SM quantum numbers which result in fractionally charged states.

We discuss the production cross-sections in Section 2 and give analytic expressions in Appendix A for general representations. There is a rich variety of phenomenologies of fractionally charged particles produced at the energy frontier depending on their quantum numbers, which we discuss roughly in Section 3, emphasizing where further dedicated theoretical or experimental study is needed to have a better handle on their signatures. In Section 4 we place bounds by reinterpreting various searches we find to be sensitive to fractionally charged particles with caveats for reasonable assumptions we have had to make as phenomenologists in the process. The constraints we find are summarized schematically in Figure 4, and the reader should be struck by the laxity of the bounds for certain combinations of quantum numbers.

Given the enormous amount these searches could teach us about the universe quite generally, it is well worth both theorists and experimentalists revisiting the possibilities for these searches, optimizing them for electric charges at least down to $e/6$, and thinking about possible new strategies for detection.

**Previous work on SM global structure**  Recent motivation for thinking about fractionally charged particles comes from discussions of the 'global structure' of the Standard Model gauge group, as we will introduce pedagogically in Section 6. The basic point is that various distinct gauge groups can nonetheless share the same structure close to the identity, which is all that is probed by minimal coupling. Nonetheless the representation theory for these different gauge groups is modified. And indeed, the Standard Model gauge group has just such an ambiguity, being

$$G_{\mathrm{SM}_n} \equiv (SU(3)_C \times SU(2)_L \times U(1)_Y)/\mathbb{Z}_n, \tag{1}$$

for $n = 1, 2, 3, 6$ (where '$\mathbb{Z}_1$' is slang for $\mathbb{1}$). We do not yet know which is realized in nature, but $G_{\mathrm{SM}_n}$ allows particles of infrared electric charge $ne/6$, and so the discovery of a particle with charge $q < e$ will distinguish between them.

The different possibilities for the global structure of the Standard Model gauge group were laid out first by Hucks [22]. The impact on the allowed line operators was studied recently by Tong [19], where it was made clear that with access to only the Standard Model degrees of freedom the different theories cannot be distinguished on flat space. The consequences of the global structure on a space of nontrivial topology have been explored in depth in [23]. Recently multiple groups have investigated how the discovery of an axion and the careful measurement of its couplings to different gauge groups also provides constraints on the global structure [24–26]. This essentially promotes the discussion in [19] about the range of the SM theta angles to a new dynamical probe—as we likewise here emphasize that a discovery of a new fractionally charged particle directly probes the allowed line operators by upgrading them to dynamical particles.

Some complementary perspectives on fractionally charged particles have recently appeared as well. In [27] the authors focus on a classification of representations consistent with general fractional charges and global structures. In particular the case where the quantum of hypercharge is smaller than expected in the SM is treated in full depth, which we will comment on only briefly below. In [28] the authors focus on the effects of fractionally charged particles in the Standard Model Effective Field Theory (SMEFT). Indeed fractionally charged particles are an interesting case of SMEFT operators being generated only at loop level, since they transform non-trivially under gauge rotations for which all SM particles are neutral, which implies that they must couple in pairs to SM matter. But resultingly the ability of SMEFT to investigate the existence of fractionally charged particles is quite limited, and we will see the energy frontier is our best probe. In some sense this is necessarily true from the generalized symmetry perspective because the emergent symmetry one finds below the mass of the lightest fractionally

Table 2: For a given representation of $SU(2)_L$ and $SU(3)_C$, fractionally charged particles are avoided only with this assignment of hypercharge. Here we list the requirements for some sample representations, but a full explanation of the structure is given in Section 6 and in particular for the Standard Model in and below Equation 25.

| $SU(3)_C$ | $SU(2)_L$ | $6Y \bmod 6$ |
|:---:|:---:|:---:|
| – | – | 0 |
| – | **2** | 3 |
| – | **3** | 0 |
| **3** | – | 4 |
| **6** | – | 2 |
| **8** | – | 0 |
| **3** | **2** | 1 |
| **6** | **2** | 5 |
| **8** | **3** | 0 |

charged particle is a global one-form symmetry under which Wilson loops are charged, but local fields are strictly blind to.[4]

## 2 LHC production

The primary phenomenological goal of this paper is to revisit collider bounds on fractionally charged particles, fleshing out their different signatures and how they are dictated by the particle's quantum numbers. The scenario we focus on is a single new Dirac fermion or complex scalar, denoted by $\psi$, sitting in an 'exotic' representation of the SM gauge group such that the electric charge of $\psi$ is some multiple $k \in \mathbb{Z}$ of $e/6$ (excluding $k = 0, \pm 6, \dots$ obviously).[5] In Table 2 we show some example non-Abelian representations and which hypercharge they *must* have to produce only integer electric charges in the far IR. Away from this choice, the electric charge will be fractional, in a multiple of $e/6$. As we will derive in Section 6, these are well-motivated to consider from the structure of the Standard Model.

We will label $\psi$ by its full quantum numbers and its electric charge when necessary, $\psi_{(SU(3),SU(2),Y),Q}$, though when the context is clear we will drop subscripts other than the charge. We denote the electric charge in fractions of $e$ throughout, e.g. using $Q = 1/3$ for $e/3$. For color singlet $\psi_Q$, the electric charge is given by the usual combination of $\tau_3$ and $Y$, while for colored $\psi$ the charge of the outgoing states is more subtle as $\psi_Q$ will combine with SM matter to form color singlet, exotically charged 'hadrons'.

We assume the only interactions $\psi_Q$ has are gauge interactions dictated by its quantum numbers. As mentioned above, interactions involving a single $\psi_Q$ and SM matter are forbidden, and we will ignore interactions between pairs of $\psi$ (really $\bar{\psi}_Q \psi_Q$, etc.) and the SM, such as $H^\dagger H \bar{\psi}_Q \psi_Q$. For fermionic $\psi$, all such interactions are non-renormalizable, while for scalar $\psi_Q$ the Higgs portal term is marginal (as is the quartic interaction $(\psi_Q^\dagger \psi_Q)^2$). Nevertheless, we will neglect this possibility as we expect it to play little role in the collider phenomenology for reasonable values of the couplings. For this initial study, we will also largely ignore the

---

[4]This 'in principle' statement is a bit too quick. There is no 'smoking gun' in SMEFT for the existence of fractionally charged particles, as some integer-charged particles can turn on all the same operators in full generality. But we anyway always must interpret some SMEFT deviation in terms of models that only add a few new particles, as you cannot directly reverse the renormalization group flow.

[5]As $\psi$ is necessarily electrically charged, it cannot be a Majorana fermion or a real scalar.

possibility of multiple exotically charged states. For certain quantum number assignments, it is possible to arrange for more renormalizable interactions between the exotic and SM sectors, such as $H\bar{\psi}_Q\psi'_{Q'}$ when one of $\psi_Q, \psi'_{Q'}$ is an $SU(2)$ doublet and $Y_{\psi'} - Y_\psi = 1/2$.[6] Multi-exotic interactions could lead to interesting phenomenology, but are beyond the scope of this paper.

Within this setup, $\psi_Q$ must be pair produced at colliders via its gauge interactions. The dominant production mechanism depends on whether or not the particles carry $SU(3)$ quantum numbers, irrespective of the spin of $\psi_Q$. For color singlets, the particles are produced in Drell-Yan $\bar{q}q \to \bar{\psi}_Q\psi_Q$ via $\hat{s}$ channel photon and $Z$. If $\psi_Q$ is an $SU(2)$ singlet, the entire cross section is proportional to $Q_\psi^2$, while the cross section for $\psi_Q$ in larger $SU(2)$ multiplets will contain pieces proportional to $(\tau_3)_\psi$, the entries of the diagonal $SU(2)$ generator appropriate for $\psi_Q$s representation. When $(\tau_3)_\psi \neq 0$, these terms typically dominate the cross section as each power of $Q_\psi$ (which we have assumed to be $< 1$) comes with a factor of $\sin^2\theta_W \sim 1/4$. For $\psi_Q$ in non-trivial $SU(2)$ representations, there is also a charged current production mode, $\bar{q}q' \to \bar{\psi}_Q\psi_{Q\pm 1} + c.c.$ via $\hat{s}$ channel $W^\pm$.

If $\psi_Q$ carries $SU(3)$ quantum numbers, QCD production $gg \to \bar{\psi}_Q\psi_Q, \bar{q}q \to \bar{\psi}_Q\psi_Q$ becomes the dominant mechanism. Of these, $gg$ is the larger channel when $\psi_Q$ is light, but $\bar{q}q$ takes over for heavier $\psi_Q$. The crossing point depends somewhat on the representation and spin of $\psi_Q$ but is $\mathcal{O}(1\,\text{TeV})$ for a Dirac fermion color triplet.

The partonic cross sections for $pp \to \bar{\psi}_Q\psi_Q$ production are compiled in Appendix A for both fermionic and scalar $\psi_Q$. For now, we opt for analytic expressions over adding new particles to Monte Carlo programs such as MadGraph [29]. In part, this is because we are focused on pair production where the expressions are still simple, but the analytic expressions also allow us to consider exotic color representations (such as a decouplet) which are not easily implemented in MadGraph. Throughout this paper we will only consider lowest order calculations, as our goal is to roughly illustrate the current bounds rather than focus on a particular search or $\psi_Q$.

Folding parton distribution functions into the partonic cross sections (Appendix A), we find the LHC proton level cross sections $pp \to \bar{\psi}\psi$. We use NNPDF3.0nlo parton distribution functions [30, 31] with $\alpha_s = 0.118$, factorization/renormalization scales of $\hat{\mu}_F = \hat{\mu}_R = \sqrt{\hat{s}}$ and assume a collider center of mass energy of 13 TeV. We have also imposed the parton-level cut $|\eta_\psi| < 2.5$ so that these particles appear in the tracker volume.

The proton level cross sections for some illustrative $\psi_Q$ are shown below in Figs. 1 and 2 below. In Fig. 1 we show the cross section for $SU(2)$ singlet $\psi_Q$, either charged only under hypercharge (left panel), or under several different color representations (right panel). Figure 2 shows the cross sections for color singlet $\psi_Q$ sitting in non-trivial $SU(2)$ representations, both via neutral current (left panel) and charged current (right panel).

The cross sections $\psi_Q$ charged only under hypercharge are quite small, $\mathcal{O}(1\,\text{pb} \times Q_\psi^2)$ for a fermionic $\psi_Q$ and $M_\psi = 100\,\text{GeV}$ and falling precipitously as $M_\psi$ increases to $\mathcal{O}(2\,\text{fb} \times Q_\psi^2)$ at $M_\psi = 500\,\text{GeV}$. Charging $\psi_Q$ under $SU(3)$, the cross section jumps by orders of magnitude, $\sigma(pp \to \bar{\psi}_Q\psi_Q) \sim 3\,\text{pb}\,(60\,\text{pb})$ for a 500 GeV color triplet fermion (color octet). The cross section for color singlet, $SU(2)$ charged $\psi_Q$ sits between these two, $\mathcal{O}(5\,\text{fb})$ for Drell-Yan production of either state in a 500 GeV doublet $\psi_Q$, and $\mathcal{O}(10\,\text{fb})$ $(\mathcal{O}(5\,\text{fb}))$ for charged current production via $W + (W^-)$. For other $SU(2)$ representations, both types of cross section grow with the size of the multiplet; labelling the $SU(2)$ part of the $\psi_Q$ state as $|I_0, i_3\rangle$, Drell-Yan $\propto i_3^2$, while the charged current is $\propto (I_0(I_0 + 1) - i_3(i_3 + 1))$. The LHC cross section for a few different $SU(2)$ multiplets (both Drell-Yan and charged current pieces) are shown in the right panel of Fig. 2.

---

[6]More exotic terms, such as $\phi_{Q'}\psi_Q f$ (where we have used $\phi_{Q'}$ for an exotic scalar in this context, $\psi_Q$ for a fermion, and $f$ a SM fermion) are also possible, either with or without flavor structure.

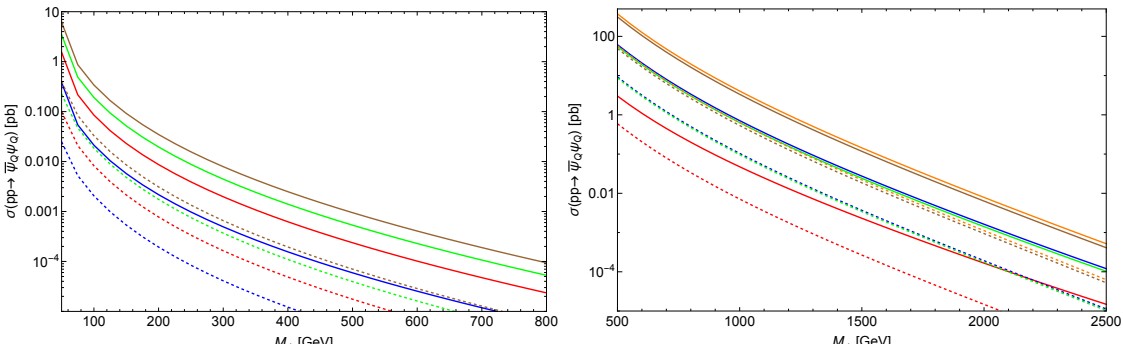

Figure 1: Left panel: Lowest order pair production cross section for $\psi_Q$ charged solely under hypercharge, $Q = 1/6$ (blue), $Q = 1/3$ (red), $Q = 1/2$ (green), $Q = 2/3$ (brown). Right panel: lowest order LHC cross section for colored $\psi_Q$ as a function of $M_\psi$ (only QCD interactions are considered). For a fixed mass, the cross section increases with the size of the representation: red (triplet), green (sextet), blue (octet), brown (decouplet) and orange (15-plet (Dynkin label (21)). In both panels we assume a center of mass energy $\sqrt{s} = 13$ TeV and use solid lines are for Dirac fermions and dashed lines for charged scalars.

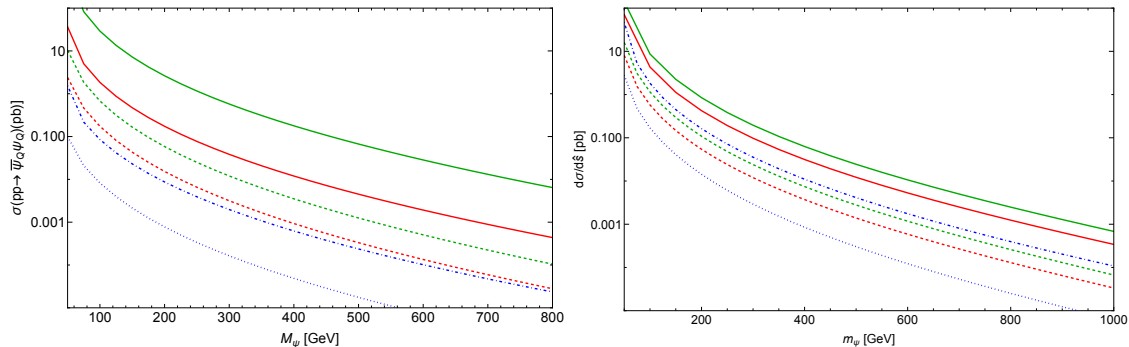

Figure 2: Cross sections for $\psi_Q$ under different $SU(2)$ representations, all with $Y = 1/3$. The red line shows the cross section for the $(\tau_3)_\psi = 1/2$ component of a $SU(2)$ doublet, while the green shows the $(\tau_3)_\psi = 1$ component of an $SU(2)$ triplet. As in Fig. 1, solid lines are for Dirac fermions while dashed are charged scalars. The blue lines (dot dashed for fermions, dotted for scalars) repeat the $SU(2)$ singlet, $Y = 1/3$ curves from Fig. 1 for comparison. Changing the hypercharge, the curves for the doublet and triplet cases would barely move, as the cross section is dominated by the $SU(2)$ portion. Right panel: Charged current cross section (via $W^+$) for doublets, triplets, and $SU(2)$ singlet for comparison.

For fixed quantum numbers, the cross sections for fermionic $\psi_Q$ are larger than their scalar counterparts by roughly an order of magnitude. This difference stems from the fact that fermions contain more degrees of freedom and that angular momentum conservation demands the amplitude to produce a pair of scalars from a pair of massless quarks/gluons is proportional to the final state velocity and therefore suppressed close to threshold.

# 3   Collider signatures of fractionally charged particles

To explore how $\psi_Q$ can be bounded at the LHC, we turn to the experiments. There are a few searches for fractionally charged particles at the LHC in the literature. The searches assume the fractionally charged particle is stable (or metastable), and rely on anomalously low $dE/dx$ in the tracking system and odd time-of-flight measurements to distinguish from background. The predominant energy loss mechanism of charged particles is via the electromagnetic interaction. For a range of quasi-relativistic velocities, this loss is described by the Bethe-Bloch equation. In this range, $dE/dx$ is independent of the particle's mass, but it is proportional to $Q^2$.

The CMS analysis [20] is the most recent and most easily translated to the scenarios we envision. In Ref. [20], events were triggered using information in the muon system, then investigated for tracks with anomalously low $dE/dx$. Events are required to have either one or two tracks, and the number of tracker hits with low ionization is used to discriminate signal from SM background. The CMS technique is optimal for $Q \sim 2/3$; particles with higher electric charge leave fewer low $dE/dx$ signals, while the analysis efficiency for lower charge states drops precipitously as lower charge leads to fewer tracker hits and therefore smaller signal/noise which inhibits track reconstruction. For $Q \simeq 1/3$, the efficiency is so poor that the bound drops to the minimum considered signal mass, 50 GeV.

A second reference we rely on is an ATLAS analysis for long lived gluinos/stops/sbottoms, Ref [21] (other searches, either for stable particles or optimized for metastable variations, can be found in Ref. [32, 33]). Upon hadronization, gluinos/stops/sbottoms all form 'R-hadrons' with integer charge, with the fraction with charge ±1 playing the largest role in the analysis. This search relies on large missing energy and/or the muon system for triggering. Given that R-hadrons are strongly interacting particles, the usage of the missing energy trigger may seem out of place. However, heavy exotic hadrons deposit negligible energy in the calorimeter, so if they are not picked up by the muon system because they are neutral (either truly neutral, as in charge zero R-hadrons, or effectively neutral for $\psi_Q$ hadrons with small $Q$), most of their energy will escape undetected. Of course, in order for this undetected energy to register as missing energy in an event, it must be balanced by something visible, either a charged exotic hadron or initial state radiation.

Regardless of how they are triggered, retained events with at least one energetic track are further scrutinized, using time-of-flight information (as determined from tracker info, muon system, or both) to separate signal from background. Because this analysis is designed for $|Q| = 1$ particles, it is not easily adapted to fractional charges much less than one. However, it is useful for estimating bounds when $Q \gtrsim 2/3$, where the CMS search loses sensitivity.

While Ref. [20, 21] are most relevant for our purposes, we'll see that $\psi$ in some corners of parameter space are best bounded by LHC searches unrelated to fractionally charged or stable particles, such as the invisible width of the $Z$ [34], monojet-style searches [35] that look for unbalanced, energetic jets, and disappearing tracks searches [36] that look for tracks which end suddenly. We will introduce more details of these searches when we encounter a scenario where they are needed.

The steps needed to go from a $pp \to \bar{\psi}_Q \psi_Q$ cross section to a bound, and exactly which bound is best, differ greatly depending on how $\psi$ is charged under the SM groups. In the next subsections, we explore some of the options.

We note also that electroweak precision observables are less constraining than collider bounds for the benchmark scenarios we consider. In large part this is because we are considering the simplest case of a single new fractionally-charged particle—with only gauge interactions, these do not contribute to $S$ or $T$, which are generally the most constraining. In a more general study of multiple fractionally-charged particles, which could include trilinear interactions with SM species, nonzero contributions could be generated. It would be interesting to

understand the constraints from precision observables on these slightly-non-minimal models to map out the full space of well-motivated fractionally charged particle signatures.

## 3.1 Solely $U(1)_Y$ charges

This is the simplest scenario, as $Q_\psi = Y$, so there is no hadronization or $SU(2)$ partners to worry about. This scenario is also the closest to the signal model used by CMS. The only difference is that CMS assumes a particle which only couples to the photon, while we include couplings both to photon and $Z$ as dictated by $Y$. As a result, we find slightly different masses corresponding to the quoted cross section limits.

## 3.2 $SU(3)_C$ charges

Colored $\psi_Q$ particles will quickly hadronize after being produced at the LHC. And if $\psi$ does not have the hypercharge demanded in Table 2, then all of the hadrons containing one $\psi_Q$ will be fractionally charged. Hadronization with the light quarks of the Standard Model will result in a variety of fractional charges for hadrons containing $\psi$. These will differ in electric charge by units of $e$, depending upon how many up-type versus down-type quarks are included.

At least as a first pass at reinterpreting the CMS search for colored representations, we follow the Lund string model [37] as used in Pythia [38] with application to R-hadrons [39]. In this model, the $\psi_Q$, $\bar\psi_Q$ sit at the endpoints of color strings which fragment. When the strings break, colored remnants join up with $\psi_Q$ to form color singlet handrons.

For color triplets, the strings break into quark-antiquark or diquark-antidiquark pairs. The three light quarks are taken to arise democratically in string breaking, modulo a phase space factor for the strange: $(u : d : s \sim 1 : 1 : 0.3)$; the diquark fraction is suppressed by an amount set by data [40, 41]. Following this model [39, 42], triplet $\psi_Q$ form mesons with $\bar u, \bar d, \bar s$ and the abundance of the 'down-type' mesons compared to 'up-type mesons' is 60:40. $\psi_Q$ baryons arise less frequently, $\sim 10\%$ of the time, with the light quark composition roughly following the same $(u : d : s)$ ratio as in $\psi_Q$ mesons.

Color octets are treated as if they connect to two strings, one giving a quark/antidiquark and the other an antiquark/diquark – which then combine with the octet to form a color singlet. The flavor composition for the gluino R-hadron case can be found in Ref. [39] and is well approximated by taking each quark/antiquark as independent and with the same $(u : d : s)$ ratio as above. For our scenario, the only difference is the charge of the hadrons will be shifted by whatever fractional charge $\psi_Q$ carries.[7]

For more exotic color representations, there is no R-hadron literature to borrow from, so we make the assumption that the bound state involving the fewest constituents are the most likely to form, and use the same $(u : d : s)$ ratio to determine the flavor (and therefore charge) of the hadrons.

The type of interpretation outlined above ignores the possibility that exotic hadrons change their electric charge via hadronic interactions as the traverse the calorimeter. For our purposes, this means that we assume the muon system triggering works out as it would in the color singlet case. Charge flipping has been modeled somewhat for R-hadrons [39, 43], which we could export to exotic color triplets or octets. However, the behavior of the bound states depends on their composition (baryonic vs. mesonic, and involving quarks vs. antiquarks), and varies depending on the phenomenological model used, so we will neglect it for this initial study. For all exotic hadrons, we ignore the mass splitting between the different exotic states and assume that the excited (higher spin) bound states immediately decay to the lowest bound state.

---

[7]Color octets can also bind with gluons (a string breaks to $gg$, with one $g$ binding to $\psi_Q$ and the other binding to remaining string fragments). Reference [39] takes to be $O(10\%)$ of $\psi$-gluon bound states, though in our case these states will retain whatever electric charge $\psi$ carries (and therefore interact with the tracker/muon system), while in the gluino case this fraction is invisible.

When using these simple hadronization rules to determine the charge of exotic hadrons, we often find some fraction of the bound states have charge $\sim 1$, e.g. 5/6 from a color triplet $\psi$ with $Y = 1/2$ (a $\psi_Q \bar{d}$ meson), or 7/6 from a color octet with $Y = 1/6$, (a $\psi u \bar{d}$). The proximity of these charges to $\pm 1$ makes the technique in CMS ineffective. To determine bounds in this scenario, we will instead reinterpret R-hadron searches from Ref [21], making the assumption that the R-hadron bounds are driven by the $Q = \pm 1$ 'meson' (i.e. $(\psi_R \bar{q})$[8] for R-hadrons from color triplet $\psi_R$ or $\psi_R q \bar{q}$ for color octet $\psi_R$) bound states and that the experiments are not sensitive to the difference between $Q \simeq \pm 1$ and $\pm 1$. For color representations not studied in R-hadron analysis, we will set bounds by equating (cross section × fraction of events with at least one exotic hadrons with near integer charge) = R-hadron cross section × fraction of events with at least one $\pm 1$ charge R-hadrons. We note that there are searches for exotic, multiply charged particles, but these searches begin at $Q = \pm 2$ [33].

This sort of reasoning will allow us to roughly reinterpret tracker based searches for some colored representations, but we emphasize that for detailed constraints dedicated simulations of hadronization and detector response for these fractionally charged representations should be done.

### 3.3 $SU(2)_L$ charges

When $\psi_Q$ sits in a non-trivial $SU(2)$ representation, it splits upon EWSB into a multiplet of $(2I + 1)$ states, for representation $I$, with components separated by $|\Delta Q| = 1$. At tree-level, and in the absence of operators such as $H^\dagger H \bar{\psi}_Q \psi_Q$ as we have assumed, the components of $\psi_Q$ are mass-degenerate. Loops of $W/Z$ bosons break this degeneracy, introducing a splitting of $\alpha_{em} m_W / \pi \sim \mathcal{O}(100)$ MeV, though with a degree of variation depending on the exact quantum numbers of $\psi_Q$. For a multiplet with hypercharge $Y$ containing a state with charge $Q = (\tau_3)_\psi + Y$ and a state with charge $Q' = (\tau'_3)_\psi + Y$ the one-loop mass difference between the two is [44, 45]:

$$M_{Q'} - M_Q = \frac{\alpha_2 M}{4\pi} \left\{ (\tau_3'^2 - \tau_3^2) \left[ f\left(\frac{m_W}{M_\psi}\right) - c_W^2 f\left(\frac{m_Z}{M_\psi}\right) \right] + 2(\tau_3' - \tau_3) Y s_W^2 f\left(\frac{m_Z}{M_\psi}\right) \right\}, \quad (2)$$

where $M_\psi$ is the tree-level mass, $c_W = \cos\theta_W, s_W = \sin\theta_W$, and

$$f(r) = \begin{cases} +r\left[2r^3 \ln r - 2r + (r^2-4)^{1/2}(r^2+2)\ln(r^2-2-r\sqrt{r^2-4})\right]/2, & \text{for a fermion,} \\ -r\left[2r^3 \ln r - kr + (r^2-4)^{3/2}\ln(r^2-2-r\sqrt{r^2-4})\right]/4, & \text{for a scalar,}[9] \end{cases}$$

In the majority of cases, the state with smaller $|Q|$ is the lightest. For $M_\psi \gg m_W, m_Z$ and using $m_Z = m_W/c_W$, we see that the mass splitting asymptotes to

$$\Delta M \simeq 160 \,\text{MeV} \times (\tau_3' - \tau_3)(\tau_3' + \tau_3 + 2Y + 2Y/\cos\theta_W). \quad (3)$$

While there can clearly be cancellations, the general trend is that the splitting grows with the hypercharge of the multiplet and the $\tau_3'$ value of the excited state.[10]

This multiplet structure has several implications for how $\psi_Q$ appears at the LHC.

- Even if one component of $\psi$ has $Q \lesssim 1/3$ – where the CMS search has limited sensitivity – it will always be accompanied by a component with larger charge. For example, a $SU(2)$ doublet with $Y = 1/3$ has one state with $Q = -1/6$, but also a state with $Q = 5/6$.

---

[8] We use a subscript $R$ for the heavy gluino/stop/sbottom in a R-hadron.

[9] The factor $k$ is UV divergent but can be absorbed by counterterms for the mass and $\psi_Q$ quartic.

[10] Note that for $Y = 0, |\tau_3'| = |\tau_3|$ the mass splitting vanishes. For $\psi$ a Weyl fermion in the $n$-dim representation, $\bar{\psi}\varepsilon^n$ transforms the same way ($\varepsilon^n$ is $n$ copies of the $SU(2)_L$ Levi-Civita), and there is an $SU(2)$ flavor symmetry between them. After $SU(2)_L$ symmetry-breaking this flavor symmetry disallows any mass splitting between the fermions of the same charge.

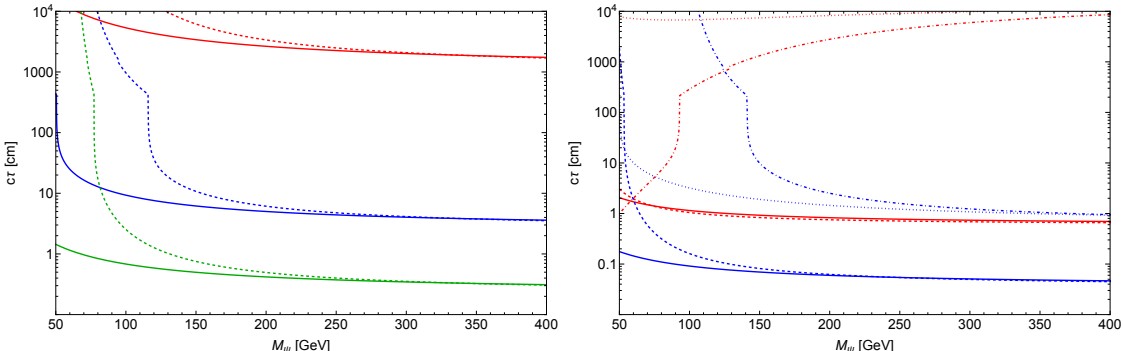

Figure 3: Decay length for the excited state(s) in an $SU(2)$ doublet $\psi$ (left panel) and $SU(2)$ triplet $\psi$ (right panel). In the left panel the blue line shows the choice $Y = 1/3$ ($Q = -1/6, Q' = 5/6$) while the green and red show $Y = 2/3$ ($Q = 1/6$, $Q' = 7/6$) and $Y = 1/6$ ($Q = -1/3, Q' = 2/3$) respectively. In all cases the $\tau_3$ component of the multiplet has the lowest (magnitude) charge. The solid lines are the results for fermionic $\psi$ while scalar $\psi$ are dashed. In the right panel, the red lines show $Y = 1/6$ while blue show $Y = 2/3$. There are more lines as there are more possible decays. The solid (dashed) red shows the decay length for $Q = 7/6$ to $Q = 1/6$ decay, while the dotted (dot-dashed) show $Q = -5/6$ to $Q = 1/6$. Unlike the case when $Y = 0$, the lifetimes of the $\tau_3 = +1$ and $\tau_3 = -1$ components are not equal. For the blue lines, the choice $Y = 2/3$ means the $\tau_3 = -1$ component has the smallest $|Q|$ and is the lightest. Therefore, the solid (dashed) lines show the decay of $Q = 5/3$ to $Q = 2/3$ while the dotted (dot-dashed) show the decay of $Q = 2/3$ to $Q = -1/3$.

- The phenomenology of the heavier, larger charge state depends crucially on its lifetime (and therefore crucially on $\psi$'s quantum numbers, which dictate the mass splitting). For mass splittings $> m_\pi$, the two-body decay $\psi_{Q+1} \to \psi_Q + \pi^+$ dominates, while for smaller splitting $\psi_{Q+1}$ mostly decays to $\psi_Q + e\,\bar{\nu}_e$ (three-body), with a small branching fraction to $\psi_Q + \mu\,\bar{\nu}_\mu$. The decay length for an illustrative set of $SU(2)$ and $Y$ choices are shown below in Fig. 3. The decay lengths asymptote at large $M_\psi/m_W$, as expected from the mass splitting formulae, while at smaller $M_\psi/m_W$ there are significant differences for fermion vs. scalar $\psi$ and cusps where the two-body decay to $\psi_Q + \pi^\pm$ turns on or off.[11]

For the selection of charges in Fig. 3, none of the excited states would be considered prompt. Several choices, such as the $Q = 7/6$, $SU(2)$ doublet state (green in the left panel of Fig. 3), or the $Q = 5/3$, $SU(2)$ triplet state (blue in the right panel of Fig. 3) have decay lengths of $O(\mathrm{cm})$ and would lead to displaced vertices or kinked tracks. A second category of excited states, such as the $Q = 2/3$ state in a $SU(2)$ doublet with $Y = 1/6$ or the $Q = 2/3$ state in an $SU(2)$ triplet with $Y = 2/3$ have accidentally small mass splitting from the lightest state in their respective multiplet, and are therefore effectively stable on collider scales. The roughly bi-modal distribution of decay lengths can be traced back to whether or not the higher charge state can decay to the lower charge state by emitting a pion.

---

[11] The one exception to the general mass splitting trend is the red dot-dashed line in the right panel of Fig. 3, the mass difference between the $Q = 2/3$ and $Q = -1/3$ components of a scalar $SU(2)$ triplet with $Y = 2/3$, which decreases for larger $M_\psi$ (leading to longer decay lengths). This is due to the fact that, while Eq. (2) generically increases the mass of the larger $|Q|$ state, there are exceptions. For example, for an $SU(2)$ triplet and $Y = 1/3$, the lightest state is the $Q = -2/3$ component rather than the $Q = 1/3$ component. The proximity of $Y = 2/3$ to $Y = 1/3$, where the 'inverted mass' situation occurs, leads to the different behavior of the mass splitting as a function of $M_\psi$.

Of course, we can have $\psi_Q$ in non-trivial representations of both $SU(3)$ and $SU(2)$, in which case the phenomenology becomes even richer, as each $SU(2)$ component will undergo hadronization, leading to a zoo of fractionally charged bound states with a variety of lifetimes.

## 4 Reinterpreted LHC bounds for assorted representations

In this section we show a sampling of the LHC bounds on different exotic $\psi_Q$ by reinterpreting a variety of searches. Given the huge number of scenarios with fractionally charged $\psi_Q$, we obviously cannot explore them all here. The goal of this benchmark study is to show roughly where things stand, identify different signal classes and detection strategies, and point out challenges and hidden assumptions in current searches.

- As our first benchmark, we take $\psi$ to be a color and $SU(2)$ singlet with $Y = Q$ a multiple of $1/6$ (obviously avoiding multiples that result in integer charge). This benchmark maps directly onto the CMS search in Ref. [20]. Using the quoted cross section numbers to bound fermionic (scalar) $\psi_Q$: $Q = 1/6$ – no LHC bound, $Q = 1/3$ $M_\psi > 88\,\text{GeV}$ ($45\,\text{GeV}$), $Q = 1/2$ $M_\psi > 610\,\text{GeV}$ ($340\,\text{GeV}$), $Q = 2/3$ $M_\psi > 650\,\text{GeV}$ ($370\,\text{GeV}$). It is worth mentioning that the bounds for the lower charge regime, $|Q| = 1/3$, have loosened substantially in Ref. [20] compared to previous iterations, Ref. [46, 47]. The loosening of the bounds can be traced to a mismodeling in the efficiency of the muon trigger for low charge [20].

For the lower charge scenarios, we must look to other searches for bounds. One obvious place to look is the invisible $Z$ partial width. If we require $\Gamma(Z \to \bar{\psi}_Q \psi_Q) \leq 1.5\,\text{MeV}$, the 1-sigma uncertainty on the invisible width [34], fermionic $\psi_Q$ with $|Q| = 1/6$ are ruled out except right at $\sim m_Z/2$ where the phase space suppression is severe. However, if relax the constraint to $2\times$ this uncertainty, the bound disappears. For scalar $\psi_Q$, $|Q| = 1/6$ there is no bound even if we impose the stronger condition of $\Gamma(Z \to \bar{\psi}_Q \psi_Q) \leq 1.5\,\text{MeV}$.

We can also approximate $|Q| \lesssim 1/3$ as invisible and constrain these scenarios using monojet style analyses $pp \to \not{E}_T + j$ [35], with the $\psi_Q$ playing the role of the missing energy. Reference [35] quotes model independent cross section limits on $pp \to \not{E}_T + j$ in bins beginning with $\sigma_{lim} < 736\,\text{fb}$ for $p_{T,j} > 200\,\text{GeV}$. Requiring such an energetic jet suppresses the cross section by $\mathcal{O}(200-500)$ depending on $M_\psi$ (larger suppression for lighter $\psi_Q$).[12,13] For fermionic $\psi_Q$, the monojet analysis places a bound of only $\sim$ few GeV, while for scalar $\psi_Q$ the cross section is so low there is no LHC bound even for massless $\psi_Q$.

Light ($\sim$ few GeV), fractionally charged $\psi_Q$ could also be similarly to millicharged matter, a topic of intense work and interest recently [48]; depending on the exact mass and charge, such scenarios are ruled out by fixed target experiments, rare meson decay, star cooling, etc. See e.g. Ref. [49, 50] for a summary of limits on millicharged matter. The most relevant bound for the range of masses and charges we are interested in comes from the SLAC anomalous single photon $e^+ e^- \to \gamma X$ search, which rules out fermionic $\psi_Q$ lighter than $10\,\text{GeV}$ for $Q > 0.08$ [51–53]. We know of no reinterpretation of this experiment in terms of a fractionally charged, complex scalar, but assume the mass bound will be in the same ballpark.

Next, let us keep the hypercharge and $SU(2)$ assignments the same but take $\psi_Q$ to be a color triplet. As we change the hypercharge assignment, we change the charge of the exotic hadrons that form, and the hadron charge determines how strict the bound is. For example:

---

[12]We derive this factor by running $pp \to \tau^+ \tau^- (+j)$ in MadGraph and varying the mass of the $\tau$.

[13]The large $p_{T,j}$ values are needed to suppress the irreducible background from $Z(\bar{\nu}\nu) + j$. The suppression this causes for our signal is much less than in dark matter models where $pp \to \not{E}_T + j$ proceeds through a contact interaction, as the latter grows with the energy.

- $Y = 0$: following the argument in Sec. 3.2 above, $\psi$ forms exotic mesons with $|Q| = 2/3$ 40% of the time, and $|Q| = 1/3$ 60% of the time. The $|Q| = 2/3$ limits from CMS are much more stringent, so equating the cross section for the production of at least one $|Q| = 2/3$ particle – $((0.4)^2 + 2 \times 0.4 \times 0.6) \times \sigma(pp \to \bar{\psi}_Q \psi_Q) = 0.64 \times \sigma(pp \to \psi\psi)$ to the CMS $|Q| = 2/3$ bound, we find masses less than 1.8 TeV (1.4 TeV) are excluded for fermionic (scalar) $\psi$. Note that $Y = 1/3$ results in hadrons with the same $|Q|$ and therefore is subject to the same bounds.

- $Y = 1/6$: For this choice, all $\psi_Q \bar{q}$ bound states have $|Q| = 1/2$. From the CMS bound, we find masses less than 1.9 TeV (1.5 TeV) are excluded for fermion (scalar) $\psi_Q$.

For our next two examples, we consider more exotic color representations, and for convenience define $d_i$ which is either a down or strange quark:

- Color octet with $Y = 1/6$: $\psi_{(8,0,1/6),1/6}$. Within our framework, this state leads to hadrons with charge $Q = 1/6 (\psi u\bar{u}, \psi \bar{d}_i d_j, \psi g)$ 55% of the time, and $Q = 7/6 (\psi \bar{d}_i u)$ or $Q = -5/6 (\psi d_i \bar{u})$ each 22% of the time. As the CMS search is insensitive to $|Q| \lesssim 1/3$ or $\sim 1$, this is a scenario where we turn to stable R-hadron searches [21] to place bounds. From this breakdown, we see that 67% of events contain at least one $|Q| \sim 1$ hadron. Equating $0.2 \times \sigma(pp \to \bar{\psi}\psi)$ to the gluino $R$-hadron cross section bound of $\sim 1$ fb, we find masses less than 2.0 TeV (1.65 TeV) are excluded for fermion (scalar) $\psi_Q$. In applying the R-hadron bounds, we are assuming the $Q = 1/6$ can be treated as neutral for the purposes of missing energy triggers.

- A color sextet with $Y = 0$: $\psi_{(6,0,0),0}$ After hadronization, this yields states with charge $Q = -4/3 (\psi u\bar{u})$, $Q = 2/3 (\psi \bar{d}_i \bar{d}_j)$ and $|Q| = 1/3 (\psi u\bar{d}_i + c.c.)$ with fractions $\sim 20\% : 30\% : 50\%$. The strongest bound comes from the $|Q| = 2/3$ fraction. The fraction of events with at least one $|Q| = 2/3$ particle is $\sim 50\%$, and equating $0.5 \times \sigma(pp \to \bar{\psi}\psi)$ to the CMS $|Q| = 2/3$ limit, we find masses less than 2.2 TeV (1.8 TeV) are excluded.

Finally, we consider benchmark color singlet $\psi$ in non-trivial $SU(2)$ representations. We pick from the examples used in the decay length plot, Fig. 3:

- An $SU(2)$ doublet with $Y = 2/3$, leading to one state with $Q = 1/6 (\psi_{(0,2,2/3),1/6})$ and one with $Q = 7/6 (\psi_{(0,2,2/3),7/6})$. The $Q = 7/6$ state decays within $\mathcal{O}(cm)$, leaving a disappearing track signature. In the context of the CMS search, the $Q = 7/6$ state just adds to the cross section for $Q = 1/6$ production, but as CMS is not sensitive to $Q = 1/6$ this gives no bound. Limits on the invisible $Z$ decay width bound $M_\psi \gtrsim 45$ GeV for either spin $\psi_Q$. Additionally, as the LHC production cross section is much larger than the $SU(2)$ singlet case, it is possible to bound this $\psi_Q$ using monojet searches. The total production for $|Q| = 1/6$ is the sum of the Drell-Yan cross sections for $|Q| = 1/6$ and $|Q| = 7/6$ along with the charged current production $pp \to \bar{\psi}_{1/6}\psi_{7/6} + c.c.$. Adding these and comparing to the 95% CL allowed cross section for $p_{T,j} > 200$ GeV, we find a monojet bound of $\sim 50$ GeV (fermionic). However, we can place a stronger bound by utilizing the disappearing track signal from the $|Q| = 7/6$ state. In a disappearing track search, events triggered with large missing energy are investigated for tracks which end, signaling the decay of a charged state into a nearly degenerate neutral state. This search strategy has been applied to the scenario of nearly degenerate higgsinos (electroweak doublets with $Y = 0$), placing a bound of 190 GeV. Applying this strategy to the scenario here, one issue is that the mass splitting between $Q = 7/6$ and $Q = 1/6$ is larger than the higgsino case. For an electroweak doublet, the mass splitting in Eq. 3 is $\propto Y$, and $Y = 2/3$ is larger than the higgsino value of $Y = 1/2$. As a result, the lifetime of the

excited state is shorter, leading to shorter tracks and a less efficient search. Taking the difference in lifetime into account and applying the cross section bound from Ref. [36], we find the current scenario is excluded for $\psi_Q$ masses below 115 GeV (70 GeV).

- An $SU(2)$ doublet with $Y = 1/6$. The only difference compared to the case above is that the states now have charge $Q = -1/3$ and $Q = 2/3$, with the $Q = 2/3$ slightly heavier. However, as $Y$ is smaller, so is the mass splitting, to the point that for $Y = 1/6$ the mass splitting drops below $m_\pi$. As a result, the lifetime of the excited state is significantly longer than in the previous case, $\mathcal{O}(20\mathrm{m})$, and we can consider it to be collider stable. We can therefore bound this scenario by ignoring the $Q = -1/3$ component and equating the total cross section for $Q = 2/3$ production, $pp \to \bar{\psi}_{2/3}\psi_{2/3} + pp \to \bar{\psi}_{-1/3}\psi_{2/3} + c.c$ to the $|Q| = 2/3$ limit from CMS [20]. We find masses below 1.1 TeV (750 GeV) are ruled out.

- An $SU(2)$ triplet with $Y = 2/3$, leading to states with $Q = -1/3, Q = 2/3, Q = 5/3$. The $Q = 5/3$ decays rapidly to the $Q = 2/3$, which then flies $\mathcal{O}(\mathrm{cm})$ before decaying to the $Q = -1/3$. Only the $Q = -1/3$ particle survives to the muon system, so if we rely on the fractionally charged bound the limits are low; summing Drell-Yan production of all three charged states along with their charged current counterparts and applying the limit from Ref. [20], we find limits of $M_\psi > 350$ GeV (200 GeV). The lifetime of the $Q = 2/3$ is long enough that one expects it should leave a trace in disappearing track searches. The limit from Ref. [36] on nearly degenerate electroweak triplets (a wino) is 650 GeV, though extrapolating this to the present scenario is not straightforward as the efficiency for the $Q = 2/3$ will be worse than the wino. Not only is the electric charge smaller, but the $Q = 2/3$ to $Q = -1/3$ mass splitting is larger (and thus its lifetime shorter) than in the charged to neutral wino case, and the sensitivity in Ref. [36] falls precipitously with mass splitting. Part of this lack in sensitivity can be compensated by a larger cross section, since we can lump the production of $Q = 5/3$ and $Q = 2/3$ together as the effective disappearing track signal. However, we find this enhancement is insufficient. The bounds fall so quickly for larger mass splittings that we estimate limits from disappearing track searches are $< 100$ GeV, worse than the fractional charge bounds relying on $|Q| = 1/3$.

The bounds from these benchmark scenarios are illustrated below in Fig. 4, and we can use our experience with those setups to extrapolate to other multiplets to some extent. For $\psi_Q$ charged solely under hypercharge, bounds come from the CMS dedicated fractionally charged search. The fractionally charged bounds are maximzed near $Q = 2/3$; for larger charge, the technique fails and is superseded by time-of-flight based searches, while for smaller charge the sensitivity drops precipitously. Monojet style searches are an interesting avenue to explore, but these perform best for heavier $\psi_Q$ – where the cross section is even lower – or contact interactions from a heavy mediator (which do not apply to our setup). For colored $\psi_Q$, the large production cross section pushes the current limits much higher, roughly 1.8 TeV for color triplets fermions. The bounds increase with the size of the $SU(3)$ representation and, at least at the level of our study, are fairly insensitive to the hypercharge of $\psi_Q$.

Comparing the above numbers we see that the scenarios we can recast into the CMS fractional charge search have slightly stronger limits than those we interpret as $R$-hadrons, as fractional charge signatures have an additional handle – low $dE/dx$ – to separate signal from background. We see the most variability in the bounds for color singlet, $SU(2)$ charged $\psi_Q$, as the signatures in the detector depend strongly on the charges and lifetimes of all the states in the multiplet. If the excited states are short lived, they add to the cross section for the lowest $|Q|$ state, but this boost can be insufficient to strongly bound the scenario if the lightest state has $|Q| \leq 1/3$. Disappearing track searches, which target the decay of the excited

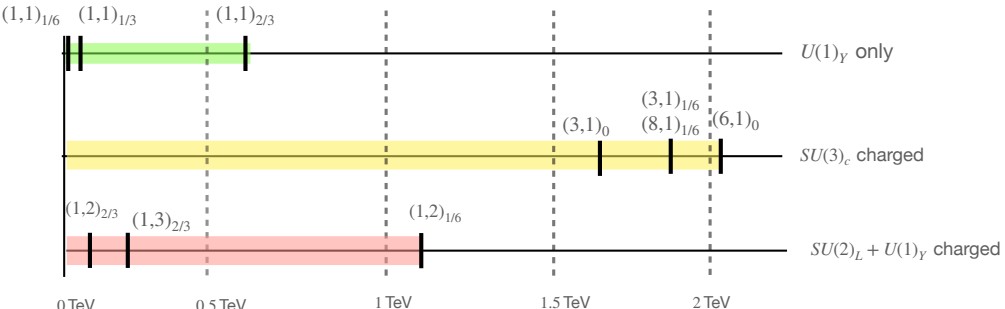

Figure 4: Graphic illustrating the mass bounds for the benchmark fractionally Dirac fermions, the details of which are discussed in the text. The bounds for fractionally charged complex scalars are lower than the fermionic case by $\sim 20\%$.

state, can provide another handle, though we find they are hampered by the fact that excited state lifetimes for fractionally charged scenarios are typically shorter than in scenarios familiar from supersymmetry (e.g. $Y = 1/2$ for pure higgsino or $Y = 0$ for wino). If the excited state happens to be long-lived, the bounds to jump significantly, as the higher charge state gives us another handle on the setup. The $SU(2)$ charged scenarios are also the most complicated, as the number of processes one needs to consider (Drell-Yan for each component, charged current between pairs of components) grows with the size of the multiplet.

We emphasize that all of these bounds are just an estimate. We have ignored higher order QCD corrections, which for inclusive cross sections are encapsulated into a $K$ factor that is typically $\sim 1 - 2$. More significantly, we have assumed that the triggering efficiency – either in the muon system efficiency or using $\not{E}_T$ – for fractionally charged particles with other (non-hypercharged) quantum numbers (or much larger mass) is not significantly different than in Ref. [20].

We conclude this section with some items worth thinking about in order to maintain a robust collider search program for fractionally charged particles.

- The LHC is an evolving apparatus, with many detector upgrades planned for the high luminosity phase. Some ways these upgrades will affect searches for fractionally charged particles include:

  - The ability to trigger using tracker information alone (at both ATLAS and CMS) may help increase sensitivity in regions where the CMS analysis is limited by the muon trigger efficiency. It is worth noting that the upgraded outer portion of the tracker will be upgraded to a digital device to facilitate the high data transfer rate needed for track triggering. However, this comes with the price that ionization energy on the individual hits is no longer kept. Multiple hits are combined together into a single output, so there will be less granular $dE/dx$ information. Exactly how much this impacts the analysis strategy for fractionally charged particles in Ref. [20] has not yet been studied.

  - The introduction of a timing layer in CMS between the tracker and ECAL will improve time-of-flight measurements, enhancing signal discrimination based on velocity or displaced vertices [54–56].

  - Reference [57] explored how the low $dE/dx$ search could be improved, especially for low $Q$, by moving from a muon trigger to a $\not{E}_T$ trigger. Detector upgrades are expected to increase the efficiency for lower $\not{E}_T$ events [58], which should help this approach further.

- The bounds above primarily rely on tracker information, using other systems only to trigger. More precise bounds, or perhaps even novel signals, could be achieved by improved modeling of the interaction of colored, fractionally charged particles as they traverse the detector. Current models are limited to heavy color triplets/octets that are lumped into hadrons with integer charge, and even within this subset there are considerable differences among models in the charge vs. neutral and meson vs. baryon fractions as a function of distance traversed [42, 43, 59].

- Some improvement in the most challenging cases is already underway from the milliQan experiment, which is forecasted to probe up to 45 GeV for a fermion of charge $e/6$ using LHC Run 3 data [60].

- Some percentage of $\psi_Q$ produced at the LHC will stop inside the detector as a result of their energy loss to the detector material. The fraction that stop depends on the mass of $\psi_Q$, its charge, and its color representation. The stopped, *stable* $\psi_Q$ may form atomic or nuclear bound states which will have a fractional charge that cannot be screened by Standard Model material. It is not clear to us whether there might be discovery potential in looking for later trajectories being subtly affected by this small persistent electric charge localized somewhere in the detector. If nothing else, it may be interesting to attempt to search disused detector parts for embedded fractional charges.

## 5 Cosmology

Not only would the discovery of a fractionally charged particle tell us an enormous amount about ultraviolet particle physics—it would also tell us a huge amount about the early universe. So for completeness we offer a brief discussion here.

Since the lightest fractionally charged particle is necessarily stable, strong constraints on the relic abundance of particles with $\mathcal{O}(1)$ electric charges are present. Our understanding thereof is mainly from the fantastic Dunsky, Hall, Harigaya papers [61, 62] as we briefly summarize in Section 5.1.

These imply that such a species could only ever have been in thermal equilibrium with the Standard Model if there were large Boltzmann factor suppression. That is, discovering a fractionally charged particle of mass $M_\psi$ gives an upper bound on the reheating temperature $T_{\text{reheat}} \lesssim M_\psi/r$. In Section 5.2 we give some basic estimates of $r$ depending on both the details of reheating and the quantum numbers of $\psi$.

This means that just as such an energy frontier discovery would falsify some of our grand models of ultraviolet physics, it would also falsify the high-scale inflation models that have been proposed in these frameworks. Of course all we know experimentally is that there was a Standard Model plasma in a radiation era at the temperatures of Big Bang Nucleosynthesis $T_{\text{reheat}} \gtrsim T_{\text{BBN}}$, but there need not have been an era of much hotter temperature [50, 63–65].

### 5.1 Abundance constraints

There have been various lab-based searches for fractionally charged particles in which a sample of some material is tested for fractional charge. Indeed the ensuing constraints on fractional charges present *in the sample* are very strong, but extrapolating to a constraint on the relic abundance is fraught with difficulties. The dust in our proto-planetary disk originated in an earlier generation of stars that underwent supernovae, and that which formed the Earth has undergone billions of years of geological activity. That is to say, tracking the evolution of heavy particles from an initial relic abundance through this non-trivial evolution requires great care. Some of these issues are discussed further in [62, 66].



However, in fact there is a better source of constraints on the relic abundance from the flux of fractionally charged particles on the Earth. In general, virialized dark matter which strongly interacts with SM particles is unable to reach underground direct detection experiments that are shielded by the Earth's atmosphere and meters of rock (see e.g. [66–68]). However, the electric charges of the states we are considering mean that there is necessarily a component which gets boosted by supernova shocks, as impressively understood in [61]. Indeed for the GeV - TeV mass range of interest at the energy frontier and the $\mathcal{O}(1)$ electric charges of our states, a relic abundance of such particles collapses into the Milky Way disk as it forms along with the baryons, thermalizes with the ISM, and undergoes Fermi acceleration from supernova shock waves. These accelerated particles appear on Earth in the form of cosmic rays, and their large boosts would allow them to penetrate the Earth down to deep underground detectors, providing strict upper limits on such a flux. In the range of parameter space of interest to us, the strictest bounds come from experiments like IceCube [69], searches for lightly ionizing particles like MAJORANA [70] and MACRO [71], and searches for magnetic monopoles like ICRR [72] and Baksan [73]. These constraints are extremely strong, giving upper bounds on the relic abundance $10^{-10} - 10^{-16}$ as a fraction of the dark matter abundance, depending on the exact charge and mass.

## 5.2 Thermal plasma production

The bounds on the relic abundance can roughly be translated into an upper bound on the reheating temperature $T_{\text{reheat}} \lesssim M_\psi / r$ where $M_\psi$ is the mass of the lightest fractionally charged particle. If we assume instantaneous reheating of all species with SM quantum numbers to a temperature $T_{\text{reheat}} \ll M_\psi$, we get a Boltzmann suppressed equilibrium abundance of $\psi_Q$:

$$n_\psi = g_\psi \left( \frac{M_\psi \, T_{\text{reheat}}}{2\pi} \right)^{3/2} \exp\left( -M_\psi / T_{\text{reheat}} \right) , \tag{4}$$

where $g_\psi$ is the number of degrees of freedom of $\psi_Q$. This gives a relic abundance relative to dark matter of

$$\frac{\Omega_\psi}{\Omega_{DM}} = \left( \frac{M_\psi \, n_\psi}{\rho_{DM}} \right) \left( \frac{s_0}{s_*} \right) , \tag{5}$$

where $s_0$ is the entropy today, $s_*$ is the entropy at $T_{\text{reheat}}$ and $\rho_{DM}$ is the average dark matter energy density. Given a bound on $\Omega_\psi / \Omega_{DM}$, we can translate Eq. 5 into a bound on $r = M_\psi / T_{\text{reheat}}$. If we impose $\Omega_\psi / \Omega_{DM} \leq 10^{-16}$, the most stringent bound in the parameter space of interest according to Ref. [61], this translates to

$$r \sim 65 , \tag{6}$$

with only weak dependence on $M_\psi$. If the $n_\psi$ produced were large we should include the effects of annihilations like for a standard freeze-out, as done in [74], but since the allowed regime is so small we can ignore this process.

Above we assumed $\psi_Q$ is instantaneously in equilibrium at $T_{\text{reheat}}$. As a test of how sensitive the $r$ value derived is to our assumptions of the reheating process, we can imagine an extreme scenario where only the SM matter is reheated at $T_{\text{reheat}}$ (which may be more or less contrived depending on the quantum numbers of $\psi$). In this case, an abundance of $\psi_Q$ is built up via freeze-in, generated from collisions among energetic SM particles on the Boltzmann tails of their equilibrium distributions. The frozen in abundance of $\psi$ can be estimated using the results of Ref. [75]. Specifically, if we assume a threshold cross section times relative velocity

of $\sigma_{SM\,SM\to\bar{\psi}_Q\psi_Q}v \sim \frac{c_{eff}}{16\pi M_\psi^2}$, where $c_{eff}$ is a combination of couplings and factors counting degrees of freedom (both initial and final), we find

$$\frac{\Omega_\psi}{\Omega_{DM}} \sim \frac{135\sqrt{5/2}\,M_{pl}\,c_{eff}\,e^{-2/r}(2/r+1)s_0}{256\,\pi^7\,g_*^{3/2}\,\rho_{DM}}\,. \tag{7}$$

For QCD production (assuming six SM fermion flavors and ignoring all SM masses), $c_{eff} \sim 75$, while production of $\psi_Q$ charged only under hypercharge has $c_{eff} \sim 0.1\,Y_\psi^2$. Plugging in numbers, the freeze-in case decreases $r$ by $\mathcal{O}(15)$ relative to the case of directly reheating $\psi$, with only some mild dependence on the value $c_{eff}$.

We note that the strong bound on $\Omega_\psi/\Omega_{DM}$ we have taken above may be loosened slightly for certain quantum numbers of $\psi$. In particular, there do exist colored representations for which all hadrons formed with SM partons have fractional electric charge, but which also have bound states with zero electric charge, such as $Q \sim (3,X)_0$ where $X = 1,3,\cdots$. Triply-exotic ($QQQ$) bound states (for $Q$ a fermion) are neutral "dark" baryons and one could investigate them as a component of DM, much as in the "colored DM" story [66, 76, 77]. However, there is a severe danger posed by the existence of mixed bounds states such as ($Q\bar{q}$) (for $\bar{q}$ a SM quark) which have fractional charges, so must have extremely suppressed relic abundances as discussed above. As understood for colored DM, the QCD phase transition automatically gives *some* suppression of the fractionally charged abundance, since $H(\Lambda_{QCD}) \ll \Lambda_{QCD}^{-1}$. Then after the QCD phase transition, many scatterings occur among the mixed bound states, which deplete their abundance in favor of the much more tightly bound ($QQQ$) by some orders of magnitude, $\Omega_{Q\bar{q}} \sim 10^{-4}\Omega_{QQQ}$. This leads to a less stringent restriction on $T_{\text{reheat}}/M_\psi$ than in a case without electrically-neutral bound states by about $\mathcal{O}(10)$.

# 6 Global structure of gauge theory

In this section we give a basic review of some group and representation theory and its appearance in gauge theories. Our focus is on conceptual understanding moreso than technical detail. The key point is to understand the differences between symmetry groups which are identical for infinitesimal symmetry transformations near the identity (they have the same Lie algebra) but differ for large symmetry transformations (they have different Lie groups as the result of non-trivial 'global structure'). This will allow us to appreciate the distinct possibilities for the gauge group of the Standard Model. Some pedagogical references for the group theory are [78, 79].

## 6.1 Abelian warmup: $\mathbb{R}$ vs. $U(1)$

Often in particle physics we are interested in continuous symmetry groups which have a notion of infinitesimal transformations which are close to the trivial, identity transformation. The earliest such example in a field theory (and indeed the farthest infrared example) is the theory of electromagnetism.

**As groups**  When we consider a gauge field theory based on a symmetry group, the gauge bosons correspond to the generators of the group. Electromagnetism has only one photon, so we are interested in groups with only one generator. In fact, the photon corresponds to the generator of $U(1)_{\text{EM}}$ gauge transformations, a global element of which we can represent as

$$U(\theta) = e^{i\theta Q}\,, \tag{8}$$

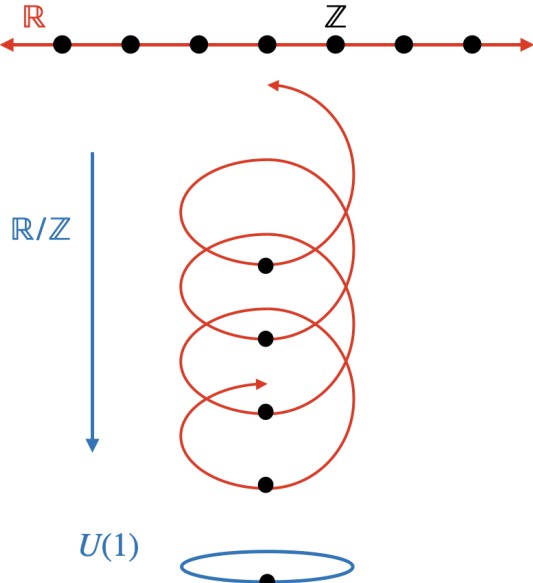

Figure 5: The group $U(1)$ constructed by quotienting $\mathbb{R}/\mathbb{Z}$. We can think about the quotient projecting the real line down to the circle such that every integer maps to the identity element.

a circle's worth of transformations which compose by complex multiplication

$$U(\theta)U(\eta) = e^{i(\theta+\eta)Q},$$

with $\theta, \eta \in [0, 2\pi)$. But alternatively we may view this as a mapping of $\theta \in \mathbb{R}$ onto the unit circle. Indeed, if we look nearby the identity transformation we cannot tell $U(1)$ from $\mathbb{R}$

$$U(\theta) \simeq 1 + i\theta Q, \tag{9}$$

where we have expanded for small $\theta$. Then we could alternatively think about just defining the group operation

$$U(\theta)U(\eta) \equiv 1 + i(\theta + \eta)Q. \tag{10}$$

This is a group which is not compact—$\theta$ has no finite period now; the group is just $\mathbb{R}$ equipped with addition. While $U(1)$ and $\mathbb{R}$ differ as Lie groups, they share the same Lie algebra.

Thinking in the other direction, if we had begun with $\mathbb{R}$ with the group operation of addition, we could see the relation to $U(1)$ by considering the quotient group $\mathbb{R}/\mathbb{Z} \simeq U(1)$. That is, we may view $U(1)$ as coming from an $\mathbb{R}$ group where we have imposed the additional equivalence relation $\theta \sim \theta + 2\pi\mathbb{Z}$—two elements of the group are now identified if they differ by an integer (the factor of $2\pi$ is a normalization convention of the period). We diagram this structure in Figure 5, and of course this is exactly what the exponential map above does.

Thinking about the physics, the perturbative, low-energy dynamics of the vector gauge bosons depend only on the gauge transformations which are close to the identity. That is, Maxwell's equations and the covariant derivative depend only on the Lie algebra of the gauge group. Yet the two theories differ in important ways, as we discuss presently.

**Electric representations:** In fact, there are nonperturbative aspects of physics which do depend on the global properties of the gauge group, and the closest at hand is simply the representation theory. In physics our objects transform in representations of the relevant symmetry groups, and the representation theory of groups with different global structures may differ.

The question in the one-dimensional case is: Which charges should be allowed? A field $\psi(x)$ with charge $q$ transforms under a $U(\theta)$ transformation as $\psi(x) \to \psi(x)\exp(iq\theta)$. If the group is $\mathbb{R}$, then any charge $q \in \mathbb{R}$ is fine. But if the gauge group is $U(1)$, then $U(2\pi) \equiv \mathbb{1}$, a rotation around the full circle is equivalent to an identity transformation. Each field must be trivially mapped back to itself by an identity transformation, but a field of general charge $q$ transforms to $\psi(x)\exp(2\pi qi)$. The requirement $\exp(2\pi qi) \equiv 1$ implies that for a $U(1)$ group we must have $q \in \mathbb{Z}$ and charge is quantized.

Thus, we see that the representation theory depends crucially on the global structure of the group, rather than just its local structure near the identity. Turned around, this means that by discovering particles with particular representations, you can learn about the global structure. If you discover two particles $\psi, \chi$ with relatively irrational charges $q_\psi/q_\chi \notin \mathbb{Q}$ then the gauge group must be $\mathbb{R}$ instead of $U(1)$. Note that you only need to discover two because for any real number can be approximated arbitrarily closely by a sequence of $aq_\psi + bq_\chi$ for $a, b \in \mathbb{Z}$.[14]

**Magnetic representations:** Gauge theories may also allow representations which carry magnetic, rather than electric charge. In the low energy theory of electromagnetism, these are the familiar Dirac monopoles. Of course it is simple enough to postulate a monopole magnetic field

$$\vec{B} = \frac{g}{4\pi}\frac{\hat{r}}{r^2} \, , \tag{11}$$

but in a quantum mechanical theory (where Aharonov-Bohm teaches us we really *must* talk about the potential $A^\mu$) such configurations connect to rich, deep physics. See e.g. Preskill's classic [81] for an in-depth introduction.

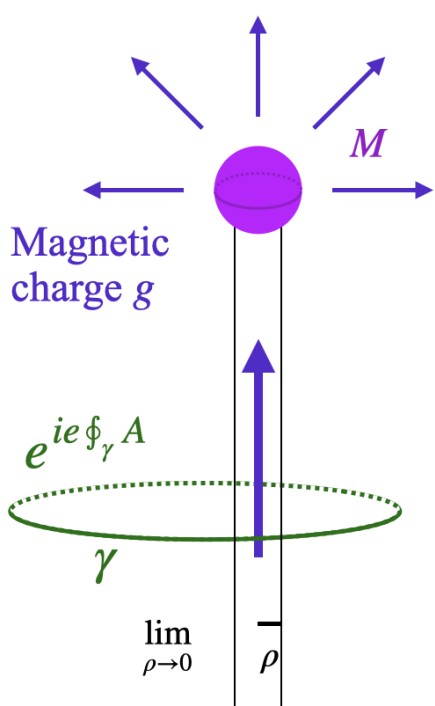

Figure 6: The Dirac monopole as the limit where a semi-infinite solenoid becomes the Dirac string.

---

[14]We note for fun that this fact was used to intriguing effect in the 'irrational axion' of [80].

The problem is that when we define the magnetic field in terms of the vector potential, $\vec{B} = \nabla \times \vec{A}$, the absence of magnetic monopoles in the Maxwell equations follows necessarily, $\nabla \cdot \vec{B} = \nabla \cdot (\nabla \times \vec{A}) \equiv 0$ because the divergence of a curl is identically zero. In the relativistic theory this is often referred to as the 'Bianchi identity', $\epsilon^{\mu\nu\rho\sigma}\partial_\nu F_{\rho\sigma} = 0$.

As Dirac understood, their construction in the low-energy theory of the gauge field $A^\mu(x)$ requires a singular line in the electromagnetic field in some direction from the monopole off to infinity known as a 'Dirac string'. This is on display in his

$$A_{\text{Dirac}}(x) = \frac{g}{4\pi r} \tan\frac{\theta}{2}\hat{\phi} \,, \tag{12}$$

in polar coordinates with $\phi$ the azimuthal angle and $\theta$ the polar angle. This indeed gives rise to the monopole magnetic field above, but this potential is singular from $r = 0$ out to all $r$ along the line $\theta = \pi$. This is not a deficiency of Dirac; any *function* $A(x)$ which produces this magnetic field will unavoidably have such a singular line, which we call a 'Dirac string'. An isolated singularity at $r = 0$ appears also of course in the electric field of an elementary charged particle—this can essentially be ignored in the low-energy theory and relativistic quantum field theory teaches us how to deal with it using renormalization. But a line-like singularity can lead to physical effects which we do not want and must avoid, as follows.

One can think of a monopole so constructed as being one end of an infinitely-thin solenoid where the other end has been sent off to infinity.[15] The magnetic flux $g$ of the monopole flows into it from infinity through the solenoid, creating a monopole magnetic field at its end. The famous Dirac quantization condition arises from requiring that the Dirac string is truly unphysical, so that we can really view the solution as just the point monopole. Given an electrically charged particle with charge $e$ and dragging it in a closed path around the would-be Dirac string of a monopole with magnetic charge $g$, the charge picks up an Aharonov-Bohm phase

$$\exp\left( ie \oint_\gamma \vec{A} \cdot \mathrm{d}\vec{s} \right) = \exp\left( ie \iint (\vec{\nabla} \times \vec{A})\mathrm{d}^2 x \right) = \exp ieg \,, \tag{13}$$

which is a physical phase we could measure in an interference experiment. Then, in order for the Dirac string to truly be unphysical, the charge $g$ of a fundamental monopole must satisfy

$$eg = 2\pi n, \quad n \in \mathbb{Z}. \tag{14}$$

The smallest-charge monopole is found for $n = \pm 1$, and of course the most stringent requirement is from the electrically-charged particle with the least charge. That is, if $q_{\text{min}}$ satisfies Eq. 14, then so will every multiple of $q_{\text{min}}$, so we have implicitly used this normalization of $e$ in writing that equation.

Alternatively to this construction (and more than 40 years later) Wu and Yang showed that magnetic monopoles can be described in a manifestly singularity-free language by using some concepts from topology [86]. In fact historically it is these ideas that have sparked theoretical physicists' enduring fascination with topology in field theory, but let us try only to appreciate some elementary points.

From this point of view, the unphysical Dirac string appears in the naive description because there is no way to express the vector potential $A^\mu(x)$ globally as a *function* for all $x$. In topological language we must instead think of fields as sections of certain fiber bundles, but elementarily we can imagine we must describe the gauge field using *two* functions $A^\mu_{\text{N/S}}(x)$ with an

---

[15]It is not clear to us who first discussed the Dirac string in this language, though Dirac's paper [82] invites this interpretation easily enough. We refer to Felsager [83] for one construction, [84] for some explicit formulae, and [85] for an experiment at creating an approximate monopole in the lab by taking just such a limit.

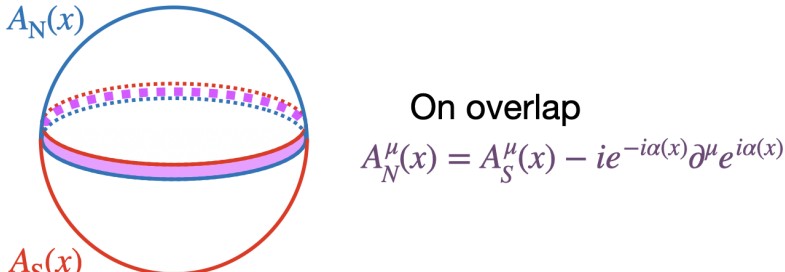

Figure 7: The local descriptions $A_{N/S}^{\mu}(x)$ of the vector potential in their separate patches, and the transition function on their overlap.

overlapping range of validity. Thinking in spherical coordinates, $A_N^{\mu}(x)$ is defined for polar angles $\theta \in [0, (\pi + \delta)/2)$ and $A_S^{\mu}(x)$ is defined on the 'southern hermisphere' $\theta \in ((\pi - \delta)/2, \pi]$ where the small $\delta$ addition to the domains ensures that these two descriptions overlap on a small ring around the equator. They have the explicit expressions

$$A_N(x) = \frac{g}{4\pi r \sin\theta} (1 - \cos\theta) \hat{\phi}, \tag{15}$$

$$A_S(x) = \frac{-g}{4\pi r \sin\theta} (1 + \cos\theta) \hat{\phi}. \tag{16}$$

If we have two overlapping descriptions on the equator they must surely somehow match, and this is possible despite them being different functions locally because there is an underlying $U(1)$ gauge redundancy. That is these functions describe the same physics on the equator if they agree up to a $U(1)$ gauge transformation, which we can see as

$$\text{On overlap:} \quad A_N^{\mu}(x) = A_S^{\mu}(x) - ie^{-i\alpha(x)} \partial^{\mu} e^{i\alpha(x)}, \quad \alpha(x) = g \frac{\phi}{2\pi} k. \tag{17}$$

Then morally speaking the different monopole solutions are classified by the value of this gauge transformation on a path around the equator $U(\phi) : \phi \to U(1)$ as $\phi = 0..2\pi$ with $U(0) = U(2\pi)$. In fact the collection of such paths is familiar in algebraic topology as the 'fundamental group' $\pi_1(G)$ of a space $G$. In the case of a $U(1)$ group, $\pi_1(G) = \mathbb{Z}$ tells us that there are magnetic monopoles labeled by any integer charge.

In contrast, in the case of an $\mathbb{R}$ gauge group there is no way to draw a closed path in $\mathbb{R}$ which cannot be shrunk down to a single point, so $\pi_1(G) = \mathbb{1}$ is trivial and this group does not have any magnetic monopoles. One may have intuited this already from the Dirac quantization condition and the results above about electric representations. Since in an $\mathbb{R}$ gauge group the electric charge can be an arbitrarily small real number, the Dirac quantization cannot be satisfied for any magnetic charges.

## 6.2 Global structure of non-Abelian groups

**Case study 1: $SU(N)$ vs. $SU(N)/\mathbb{Z}_N$**    Recall that the group $SU(N)$ consists of $N \times N$ complex matrices which are unitary ($V^{\dagger}V = 1$) and special ($\det V = 1$). The structure of infinitesimal transformations in $SU(N)$ is generated by traceless hermitian $N \times N$ matrices

$$U(\theta^a) = \mathbb{1}_j^i + i\theta^a (T^a)_j^i, \tag{18}$$

where $a = 1..N^2 - 1$. These $T^a$ generate the Lie algebra of $SU(N)$ in a way that generalizes the familiar Pauli matrices of $SU(2)$. The group $SU(N)$ is non-Abelian but it has a nontrivial

'center' $\mathbb{Z}_N$, where the center of a group is the subgroup of elements which commute with all others,

$$\mathbb{Z}_N \subset SU(N) : \left\{ \exp\left(\frac{2\pi k}{N} i\right) \mathbb{1}_N \right\}_{k=0..N-1}, \tag{19}$$

which is generated by the element $\omega_N = \exp\left(\frac{2\pi}{N} i\right) \mathbb{1}_N$. We can sensibly form the quotient group $SU(N)/\mathbb{Z}_N$ where we 'mod out' by the center subgroup. This group can be thought of as $SU(N)$ with the equivalence relation $\omega_N \sim \mathbb{1}_N$ imposed. But this does not change the structure of transformations near the identity; the Lie algebra remains the same.

In the quotient group any two elements of $SU(N)$ which differ by a center element are now identified. In particular, each element of the center is now identical to $\mathbb{1}_N$. Thinking now about the representation theory, this means that such elements must necessarily act trivially on each field.

If we think about the familiar $SU(N)$ representations, this is not the case for all of them. Consider a field $\psi^a$ in the fundamental representation of $SU(N)$, which transforms generally as $\psi^a \to \psi^a V_a^b$. Then in particular under an $(\omega_N)_a^b$ transformation it picks up an $N^{\text{th}}$ root of unity phase. In $SU(N)$ this is as it should be, but this is nonsensical for a representation of $SU(N)/\mathbb{Z}_N$, in which this element was literally the identity—then the fundamental representation of $SU(N)$ is not an allowed representation of $SU(N)/\mathbb{Z}_N$!

The field theory of $SU(N)/\mathbb{Z}_N$ is a theory of *adjoint* fields, including of course the gauge bosons which are necessarily present. An adjoint representation can be thought of as the product of a fundamental and antifundamental with the trace removed, with the math $N \otimes \bar{N} = (N^2 - 1) \oplus 1$. With equal number of fundamental and antifundamental indices, $A_c^a \to (V^\dagger)_d^c A_c^a (V)_a^b$ is easily seen to be invariant under a center transformation. The $SU(N)/\mathbb{Z}_N$ theory allows arbitrary matter which is in either the adjoint or irreps which can be built from it and the Levi-Civita symbol $\varepsilon_{a_1...a_n}$.

The global structure also here crucially changes the topological properties of the gauge group, just as did the quotient in the Abelian case. We can see this again in the allowed magnetic representations, which are controlled by the fundamental group $\pi_1(G)$. This can be thought of elementarily as simply the group of topologically equivalent maps of circles into $G$, $\pi_1(G) \simeq \{\phi : S^1 \to G\}$. The question is what sorts of closed loops we can draw in $G$. For $SU(N)$ it is a fact that $\pi_1(SU(N)) = 1$ and there are no magnetic monopoles. But now let us consider the following diagonal generator of $SU(N)$

$$T^{N^2-1} = \begin{pmatrix} 1 & & & \\ & 1 & & \\ & & \ddots & \\ & & & -(N-1) \end{pmatrix}, \tag{20}$$

which is a hermitian, traceless matrix you can think of as the generalization of the Pauli $\sigma_3$ to $SU(N)$. Of course close to the identity we can think of an infinitesimal transformation in this direction $\theta^a = \delta_{N^2-1}^a \theta$,

$$U(\theta^a) = \mathbb{1} + i\theta T^{N^2-1} + \mathcal{O}(\theta^2), \tag{21}$$

just as in $SU(N)$. But now in $SU(N)/\mathbb{Z}_N$ we will see something interesting if we go a *large* distance in this direction, say $\theta = 2\pi/N$. The higher order terms form into the exponential

$$U(\theta^a) = \exp\left(i\frac{2\pi}{N} T^{N^2-1}\right) = \exp\left(i\frac{2\pi}{N}\right) \mathbb{1}, \tag{22}$$

and because $-(N-1) = 1 \pmod N$ we see that by following a path along the $T^{N^2-1}$ direction we have ended up at an element of the $\mathbb{Z}_N$ center. In $SU(N)$ there's nothing special to say

about this, but in $SU(N)/\mathbb{Z}_N$ this means that you can go far out along this direction and end up *back at the origin*! So now there is a map $\phi : [0, 2\pi) \mapsto G$ where $\phi(\theta) = U(\theta/N)$ and this gives us one-dimensional loops around $SU(N)/\mathbb{Z}_N$.

This means that in addition to the electric representations discussed above, $SU(N)/\mathbb{Z}_N$ also has magnetic representations. In this case there are not monopoles of any integer charge as in $\pi_1(U(1)) = \mathbb{Z}$ but rather only $N$ distinct closed loops $\pi_1(SU(N)/\mathbb{Z}_N) = \mathbb{Z}_N$ and so only $N$ distinct monopoles. If you wind $N$ times around $SU(N)/\mathbb{Z}$ you end up with a path that can be deformed into lying only in $SU(N)$, where it can be shrunk to a point.

The familiar example of this is $SU(2)$ which has $\pi_1(SU(2)) = 1$, and you will recall is only locally isomorphic to the rotation group $SO(3)$, while globally double-covering it. Then the quotient group $SU(2)/\mathbb{Z}_2 \cong SO(3)$ is isomorphic to 3D rotations and has $\pi_1(SU(2)/\mathbb{Z}_2) = \pi_1(SO(3)) = \mathbb{Z}_2$. The fact that looping $N$ times around $SU(N)/\mathbb{Z}_N$ returns you to the identity is nothing more than 'Dirac's belt trick'—in 3D space taking the belt buckle on a loop in $\phi : [0, 2\pi) \mapsto SO(3)$ puts it in a topologically twisted sector yet going around twice returns it to the identity.

**Case study 2: $SU(N) \times U(1)$ vs. $U(N)$**   In the case of a product group there may be a more subtle choice of global structure which interrelates the allowed representations of the group factors. In fact $U(N) \cong (SU(N) \times U(1))/\mathbb{Z}_N$ differs in its global structure from $SU(N) \times U(1)$, though the fact that they are equivalent locally is often used when analyzing perturbative physics.

In this case the quotienting is done by a diagonal combination of the $\mathbb{Z}_N$ center subgroups of the two factors, and identifies them with each other $\exp\frac{2\pi i}{N}\mathbb{1}_N \sim \exp\frac{2\pi i}{N}Q$. This means that every field must be invariant under the diagonal combination of rotations, $\exp\frac{2\pi i}{N}\mathbb{1}_N \times \exp\frac{-2\pi i}{N}Q \equiv 1$.

There is in general for $SU(N)$ representations a notion of '$N$-ality' which simply tracks how the field transforms under a $\mathbb{Z}_N$ center transformation. A fundamental has $N$-ality of 1, as we saw above, and in the $SU(N)/\mathbb{Z}_N$ theory the representation theory required $N$-ality of 0 (mod $N$). Here in the $U(N)$ theory the quotient instead correlates the $N$-ality of the representations with the Abelian charge. A fundamental must have a charge under $Q$ which is 1 (mod $N$) such that it is invariant under the quotiented subgroup. Since every representation may be constructed by taking tensor products of fundamental and anti-fundamental representations, this informs us of the charge $Q$ (mod $N$) which each $SU(N)$ representation must have in order to be an allowed representation of $(SU(N) \times U(1))/\mathbb{Z}_N$. The two-index $\phi^{ab}$ either symmetric or anti-symmetric irrep comes from $N \otimes N = N(N-1)/2 \oplus N(N+1)/2$ so must have $U(1)$ charge 2 (mod $N$). The adjoint $\phi^a_b$ is built from $N \otimes \bar{N} = N^2 - 1 \oplus 1$ so must have $U(1)$ charge 0 (mod $N$), and so on.

Now what of the magnetic representations? Early physics work in this direction includes [10, 87–90], in which much further detail may be found. In $SU(N) \times U(1)$ the two factors are separate, and $\pi_1(SU(N)) = 1$ does not have monopoles while $\pi_1(U(1)) = \mathbb{Z}$ gives the simple monopoles familar from the Abelian case above.

Turning to $(SU(N) \times U(1))/\mathbb{Z}_N$, the structure is a bit subtle. The fundamental group $\pi_1(U(N)) = \mathbb{Z}$ tells us we have distinct monopoles for any integer, but in this case the spectrum of monopoles is skewed away from just being the $\mathbb{Z}$-valued monopoles of the Abelian group. Let us picture the different classes of closed paths. Of course one thing we can do is simply go all the way around $U(1)$ as $U(\phi) = \exp(i\phi Q)$ and wrap around the $U(1)$ direction to get a monopole with only $U(1)$ magnetic flux.

However, now if we go a fraction of $k/N$ around the circle, the quotient combined with our understanding of the $SU(N)/\mathbb{Z}_N$ case above tells us $\exp\left(i\frac{2\pi k}{N}T^{N^2-1}\right) \sim \exp\left(i\frac{2\pi k}{N}Q\right)$. Then we can return to the origin not by continuing around the $U(1)$ direction, but by taking a path

Table 3: Representations of the Standard Model fields under the subgroups of the gauge symmetries, switching notation from the earlier sections in which we used Dirac fermions and the standard convention for the normalization of hypercharge. Herein we speak of Weyl fermions—as appropriate for the Standard Model in the unbroken phase—and henceforth we normalize $U(1)_Y$ so the least-charged particle has unit charge. This will make various statements simpler to see.

|          | $Q_i$ | $\overline{u}_i$ | $\overline{d}_i$ | $L_i$ | $\overline{e}_i$ | $H$ |
|----------|-------|------------------|------------------|-------|------------------|-----|
| $SU(3)_C$ | **3** | $\overline{\mathbf{3}}$ | $\overline{\mathbf{3}}$ | – | – | – |
| $SU(2)_L$ | **2** | – | – | **2** | – | **2** |
| $U(1)_Y$ | $+1$ | $-4$ | $+2$ | $-3$ | $+6$ | $-3$ |

along $T^{N^1-1}$ in $SU(N)$ that when we get close to the origin looks like $U(\theta) = \mathbb{1} + i\theta\, T^{N^2-1}$.

So this case is something of a mixture of the two we have seen before. There are $k \in \mathbb{Z}$ magnetic monopoles, but they now have both Abelian and non-Abelian magnetic fluxes for $k \neq 0 \pmod{N}$. It is only in the case $k \in N\mathbb{Z}$ for which they have $U(1)$ magnetic flux only.

## 6.3 The Standard Models

The case of the Standard Model is not much more difficult than the above examples we have discussed. As you know, the Standard Model is a Yang-Mills theory with a certain continuous gauge group which near the identity includes factors of $SU(3)_C$, $SU(2)_L$, and $U(1)_Y$. The perturbative physics of these theories, including the spectrum of gauge bosons, is controlled by the local structure of gauge transformations which are close to the identity transformation. Thinking just of the symmetry group, we may write a general such infinitesimal group element as

$$U(\theta_1, \theta_2^i, \theta_3^a) = \mathbb{1} + i\theta_1 Y + i\theta_2^i T_2^i + i\theta_3^a T_3^a, \tag{23}$$

where $\theta_{1,2,3}$ parametrize the transformations in the hypercharge, weak, and strong directions, and $T_3, T_2, Y$ are the generators of the respective subalgebra. Thinking about the SM as a Yang-Mills theory we wish to upgrade this invariance from global to local transformations which depend on spacetime position $\theta_i \mapsto \theta_i(x)$. Then as is familiar we must introduce vector gauge bosons in the adjoint representation and couple them to our charged fields.

The transformations close to the identity explore only the Lie algebraic structure, and in fact are not sensitive to the 'global structure' of the gauge group. This is what we see in the covariant derivative to minimally couple charged particles to a gauge field

$$D_\mu = \partial_\mu - i g_1 Q_Y B_\mu - i g_2 T_{R_2}^\alpha W_\mu^\alpha - i g_3 T_{R_3}^a G_\mu^a, \tag{24}$$

which explores only the local structure of the gauge group, just as the position derivative explores only the local structure of the spacetime manifold. That means we are only experimentally sure of this local information, and in fact there are multiple possible Lie groups which have this same Lie algebra.

The four different possibilities are

$$G_{\mathrm{SM}_n} \equiv (SU(3)_C \times SU(2)_L \times U(1)_Y)/\mathbb{Z}_n, \tag{25}$$

where $n = 1, 2, 3, 6$ and we use the slang term $\mathbb{Z}_1 \equiv \mathbb{1}$ for convenience. As far as we are aware, this was first laid out systematically in a little-known 1990 solo paper by a UCSB grad student [22] but has been well-publicized in recent years [19]. The options with $n > 1$ can be viewed

Table 4: How each SM field transforms under a center transformation by the generator of each noted subgroup.

|  | $Q_i$ | $\overline{u}_i$ | $\overline{d}_i$ | $L_i$ | $\overline{e}_i$ | $H$ |
|---|---|---|---|---|---|---|
| $\mathbb{Z}_3 \subset SU(3)_C$ | $e^{i2\pi/3}$ | $e^{-i2\pi/3}$ | $e^{-i2\pi/3}$ | 1 | 1 | 1 |
| $\mathbb{Z}_2 \subset SU(2)_L$ | $-1$ | 1 | 1 | $-1$ | 1 | $-1$ |
| $\mathbb{Z}_3 \subset U(1)_Y$ | $e^{i2\pi/3}$ | $e^{-i2\pi/3}$ | $e^{-i2\pi/3}$ | 1 | 1 | 1 |
| $\mathbb{Z}_2 \subset U(1)_Y$ | $-1$ | 1 | 1 | $-1$ | 1 | $-1$ |

as quotient groups of $G_{\text{SM}_1}$ where we quotient out certain diagonal center transformations as follows.

In the group $G_{\text{SM}_2} = G_{\text{SM}_1}/\mathbb{Z}_2$, we impose an equivalence relationship between the $\mathbb{Z}_2$ center subgroups of $SU(2)_L$ and of $U(1)_Y$. That is, $(-1)\mathbb{1}_2 \sim \exp(i\pi Y)$, working now in the normalization that the least-hypercharged particle has unit charge (see Table 3). In the group $G_{\text{SM}_3} = G_{\text{SM}_1}/\mathbb{Z}_3$, we impose an equivalence relationship between the $\mathbb{Z}_3$ center subgroups of $SU(3)_C$ and of $U(1)_Y$. That is, $\exp(2\pi i/3)\mathbb{1}_3 \sim \exp(i2\pi Y/3)$. In the group $G_{\text{SM}_6}$ we impose both of these quotients simultaneously.

Of course we can always consider these as abstract quotient groups, as in the constructions of the previous sections. But we have also observed the particles of the SM, which transform in a variety of representations. To see if we can legitimately consider these other possibility for global structure, we must check that the representation theory of any of these options actually allows for the needed particles.[16] Indeed, it does work, as may be checked easily from the data in Table 3. In the case of the $\mathbb{Z}_2$ quotient, we see that the fields which are $SU(2)$ doublets all have odd hypercharge, and the fields which are $SU(2)$ singlets all have even hypercharge (and the $SU(2)$ triplet $W^a$ of course has zero hypercharge) which means that indeed none of the fields are charged under this diagonal $\mathbb{Z}_2$ center transformation. The $\mathbb{Z}_3$ subgroup may be checked just as easily and the conclusion is the same, meaning that indeed there is a four-fold ambiguity in the global structure of the gauge group of the SM.

It is useful also to note that a particular global structure may be demanded by the UV embedding of the SM in a unified gauge group. Either of $SO(10)$ or $SU(5)$ demand the $\mathbb{Z}_6$ quotient. Less stringently, Pati-Salam $SU(4)_C \times SU(2)_L \times SU(2)_R$ requires the $\mathbb{Z}_3$ quotient and trinification $SU(3)_C \times SU(3)_L \times SU(3)_R$ needs the $\mathbb{Z}_2$ quotient.

Given an embedding of the SM gauge algebra in a UV theory, we can see the global structure demanded simply by examining the decomposition of the fundamental irreps of the UV under this breaking, and asking which center elements they are invariant under. For example, the embedding of the SM in $SU(5)$ is such that the fundamental decomposes as $5 \rightarrow (3,1)_{+2} \oplus (1,2)_{-3}$, and we see manifestly that these are invariant under the $\mathbb{Z}_6$ center. Since all irreps of $SU(5)$ can be found in tensor products of 5 and $\bar{5}$, the embedding of the SM in $SU(5)$ produces only representations which are invariant under the $\mathbb{Z}_6$. More formally, of course, one can find group theoretically that it really is $SU(3) \times SU(2) \times U(1)/\mathbb{Z}_6$ which is actually a subgroup of $SU(5)$, as has been known since 1980 at latest [97].

---

[16]One must additionally check that each of these versions of the Standard Model is free of global anomalies, which is indeed true as discussed in [91–96].

From the above argument, it is clear that finding a representation which is charged under the $\mathbb{Z}_6$ center falsifies the embedding into $SU(5)$. More generally, discovering a particle with electric charge $e/6$ (either at colliders or elsewhere) would rule out all the minimal unified models of the universe.[17] A new particle with charge $e/2$ would tell us we can have Pati-Salam but it cannot be further embedded into $SO(10)$, and a new particle of charge $e/3$ would allow a unified theory like trinification but rule out its embedding in $E_6$.

**Some additional possibilities:**    Thinking just as low-energy effective field theorists, there are a couple further possibilities are useful to note. For one, it is conceivable that the hypercharge assignments we have in Table 3 are not actually in terms of the charge quantum. That is, we could discover a particle which has hypercharge $1/N$ that of the left-handed quark doublet field $Q$. This would rule out all the UV unification models we normally think about, but is possible. In terms of thinking about the global structure of the SM gauge group, this would effectively tell us that the $U(1)_Y$ circle is actually a factor of $N$ 'larger' than we had thought. Correspondingly the magnetic monopole charges are a factor of $N$ larger as a result of Dirac quantization. Recently [27] has fully classified which such possibilities are consistent with the various SM quotients. It would be interesting to understand which of these could still be consistent with new unification models.

Most exotically, we can think about $\mathbb{R}_Y$, in which irrrational charges are allowed. At a generic point in some constraint plot of fractionally charged particles, one can have this in mind as the alternate hypothesis that is being tested. It is true that we expect theories of quantum gravity do not contain non-compact gauge groups like $\mathbb{R}$ (see e.g. [15, 100]), but it is not obvious there is anything wrong with them strictly as quantum field theories. Flipped around, we can say that searches for irrationally fractionally charged particles are testing deep principles of UV physics. These ideas are also subject to precision tests of atom neutrality for example using interferometry [101–103].[18]

Finally we mention that these are not the only possible ambiguities in the gauge group of the Standard Model. In [105] (App. B.1) we introduced the SM$^+$, in which the SM is extended by gauging $\mathbb{Z}_{N_c N_g}^{B-N_c L} \times \mathbb{Z}_{N_g}^{L}$, which is the Standard Model's anomaly-free, generation-independent, global zero-form symmetry. This entails no modification of the local dynamics, but ensures absolute proton stability. We will further explore these and related possibilities in future work [106].

# 7   Generalized Global Symmetries

As fundamental physicists we are deeply familiar with the power of symmetries and how a proper understanding of the symmetries of a system can aid in both our description of the theory and in finding a further ultraviolet description thereof. In the previous section we discussed the ambiguity in the global structure of the Standard Model gauge group as general, bottom-up motivation for searching for fractionally charged particles. Focusing on a limit perturbatively close to the free theory, this may seem like just such a symmetry analysis, but

---

[17]Notably this statement only applies for the minimal theories of so-called 'vertical' unification; that is theories which consolidate one generation of SM fermions into fewer irreps. Unification among generations may be compatible with the existence of any of these fractionally charged particles. Obviously so when the horizontal gauge group is factorized from the Standard Model gauge group e.g. [98], but even in non-factorized cases such as color-flavor unification [99].

[18]Of course all experimental measurements have a finite precision, so in some strict sense it is not possible to 'prove' an electric charge to be irrational. Regardless, even measuring a rational charge with very large denominator (when expressed irreducibly) would be challenging to UV physics, as it has proven hard to find large charges in string theory [104].

*really* gauge 'symmetries' are *not* symmetries—they are redundancies of our description. This is evident in the existence of descriptions where we never need speak of a gauge redundancy, such as the 'on-shell approach', and has been hammered home to us by discovering dualities where the same physics can be understood in terms of gauge theories with different groups. So it is useful instead to focus on global symmetries, which do have physical content that is independent of any choice of description. In this section we discuss how the possible global symmetries of the Standard Model also provide a bottom-up motivation for the search for fractionally charged particles.

In the framework of Generalized Global Symmetries, symmetries correspond to the existence of certain operators which have topological correlation functions. These are known as 'symmetry defect operators' (SDOs), can be thought of as implementing the global symmetry transformation by 'acting on' (or 'sweeping past') the charged objects, and beautifully generalize familiar notions like Noether charges and Gauss' law.

In the following we will aim to describe relevant basic ideas of Generalized Global Symmetries in an elementary fashion intending to convey some conceptual lessons. For further detail, generalization, and technicalities we refer to the seminal [17] and to some reviews aimed to be accessible for particle physicists [107–109].[19] But we will eschew any topic whose introduction would require cohomology, as well as many interesting GGS possibilities broader than the basics we require. Ideas and technology from GGS are gradually being utilized in (or towards) particle physics applications, for example [23–26, 91, 92, 95, 96, 98, 99, 117–146].

**Familiar (zero-form) Noether charges**    A familiar symmetry which acts on local fields (so the charged operators are zero-dimensional) has an associated Noether charge. In the case of a continuous symmetry (for simplicity, $U(1)_X$) we may build this out of a Noether current $J^\mu$ which obeys the conservation equation $\partial_\mu J^\mu = 0$. From this current we can build a family of topological, unitary operators by exponentiating its integral over any three-manifold $\Sigma_3$,

$$U_\alpha(\Sigma_3) = \exp\left( i\alpha \int_{\Sigma_3} J^\mu \epsilon_{\mu\nu\rho\sigma} dx^\nu dx^\rho dx^\sigma \right), \tag{26}$$

where $\epsilon_{\mu\nu\rho\sigma} J^\mu \equiv \star J$ is the Hodge dual. We refrain from the index-free notation of differential forms, but mention that the benefit thereof is to emphasize that the metric tensor is not needed to define these operators—they are supposed to be topological, after all.

The familiar Noether charge restricts $\Sigma_3$ to be all of space at a given time, and the topological invariance of the charge is then the fact that it can be moved forward or back in time and the charge remains the same. But this more covariant set of operators is well-defined for any $\Sigma_3$, and the conservation $\partial_\mu J^\mu = 0$ implies that any deformations of this surface do not change the correlation functions of $U_\alpha(\Sigma_3)$. Let us discuss further how to think about this, drawing from [107] among others.

We consider smoothly deforming $\Sigma_3$ to $\Sigma_3'$, where for now we assume doing so does not cross any charged operators. That is, the spacetime volume in between these is a four-manifold $\Sigma_4$ bounded by these two three-surfaces, $\partial\Sigma_4 = \Sigma_3 \bigcup \Sigma_3'$, and $\Sigma_4$ does not have any charged operators in it. We compute the product of an SDO on $\Sigma_3$ implementing a rotation by $\alpha$ and an SDO on $\Sigma_3'$ implementing a rotation by $-\alpha$ using the generalized Stokes' theorem

$$U_\alpha(\Sigma_3)U_{-\alpha}(\Sigma_3') = \exp\left( i\alpha \int_{\Sigma_3} J^\mu \epsilon_{\mu\nu\rho\sigma} dx^\nu dx^\rho dx^\sigma - i\alpha \int_{\Sigma_3'} J^\mu \epsilon_{\mu\nu\rho\sigma} dx^\nu dx^\rho dx^\sigma \right) \tag{27}$$

$$= \exp\left( i\alpha \int_{\Sigma_4} \partial_\mu J^\mu d^4x \right) = 1. \tag{28}$$

---

[19]We note also introductions and reviews a bit further afield such as [110–116].

Where we have used current conservation to find the volume integral vanishes and we get 1 on the right-hand side. Since these SDOs are unitary operators, we learn $U_\alpha(\Sigma_3) \simeq U_\alpha(\Sigma_3')$. That is correlation functions containing an insertion of $U_\alpha(\Sigma_3)$ are invariant under deforming $\Sigma_3$, so the SDOs are topological as we said above.

Now, the above equations assumed that there are no charged particle in the volume $\Sigma_4$ between the initial and final surfaces. How do the SDOs behave when we move the surface $\Sigma_3$ past a local field $\psi(y)$ charged under $U(1)_X$?

Recall that the Ward identity encodes how the conservation of a symmetry current jibes with the existence of operators sourcing that current. That is, we must upgrade the classical $\partial_\mu J^\mu(x) = 0$ to an operator equation which tells us what to do with a charged field $\psi(y)$. One derives the consequences of the symmetry in the quantum mechanical theory by performing a symmetry transformation for a general correlation function calculated by a path integral, demanding the action is invariant under the symmetry, and observing the consequences for the charged operators—for example in Section 14.8 of Schwartz [147]. In the Abelian case we have simply

$$\partial_\mu J^\mu(x)\psi(y) = \delta^{(4)}(x-y)q_\psi\psi(y). \tag{29}$$

This tells us that while $\partial_\mu J^\mu(x) = 0$ away from other operators, there are important contact terms when this symmetry current hits an operator charged under this symmetry. One should properly view such statements as taking place inside arbitrary correlation functions separated from other local operators,

$$\langle\dots\partial_\mu J^\mu(x)\psi(y)\dots\rangle = \delta^{(4)}(x-y)q_\psi\langle\dots\psi(y)\dots\rangle, \tag{30}$$

where the '$\dots$' is a stand-in for any other operators away from $x, y$. The action of the Ward identity will be crucial in understanding the use of the symmetry defect operators.

Now let us repeat the computation above of deforming $\Sigma_3$ to $\Sigma_3'$ but now in the case where doing so *does* cross a charged operator. A simple case has $\Sigma_3$ as a hypersphere $S^3$ and the local operator $\psi(x)$ at a point $x$ which is inside $\Sigma_3$. We consider then shrinking $\Sigma_3$ down $\Sigma_3 \to \Sigma_3'$ so $x$ is now outside of this surface, as in Figure 8, and then acting with the inverse SDO. Overall this acts on $\psi(x)$ as

$$U_\alpha(\Sigma_3)\psi(x)U_{-\alpha}(\Sigma_3') = \exp\left(i\alpha\int_{\Sigma_4}\partial_\mu J^\mu d^4x\right)\psi(x) = \psi(x)e^{i\alpha q_\psi}. \tag{31}$$

Where we have used the Ward identity and the fact that $x \in \Sigma_4$, and we refer to [107] for further detail. We note also that if no other charged operators were in $\Sigma_3$ to begin with, then conceptually we can skip this second step of acting with $U_{-\alpha}(\Sigma_3')$ and just imagine shrinking $\Sigma_3$ all the way down to a point after it passes $x$.

We can state the result more generally by saying that these SDOs act by 'linking', and writing $U_\alpha(\Sigma_3)\psi(x)U_{-\alpha}(\Sigma_3') = \psi(x)e^{i\alpha q_\psi \mathrm{Link}(\Sigma_3,x)}$. In the situation we have described, the 'Linking number' $\mathrm{Link}(\Sigma_3,x)) = 1$. The 'Linking number' is a topological invariant of a configuration in $d$ spacetime dimensions between a $p$-dim submanifold $\Sigma_p$ and a $d-p-1$-dim submanifold $\Sigma_{d-p-1}$. This action by linking keeps track of the charge inside the SDO when we move a charged operator from the interior to the exterior or vice-versa. To gain some intuition, it is useful to think about the case $d = 3$ (say, 3-space at some fixed time), where it's easy to visualize that a $p = 0$ point is either inside or outside a $d-p-1 = 2$ sphere, and a $p = 1$ loop can be linked with another $d-p-1 = 1$ loop.[20]

---

[20]We note for fun that general linking numbers can be defined by certain topological quantum field theories [148].

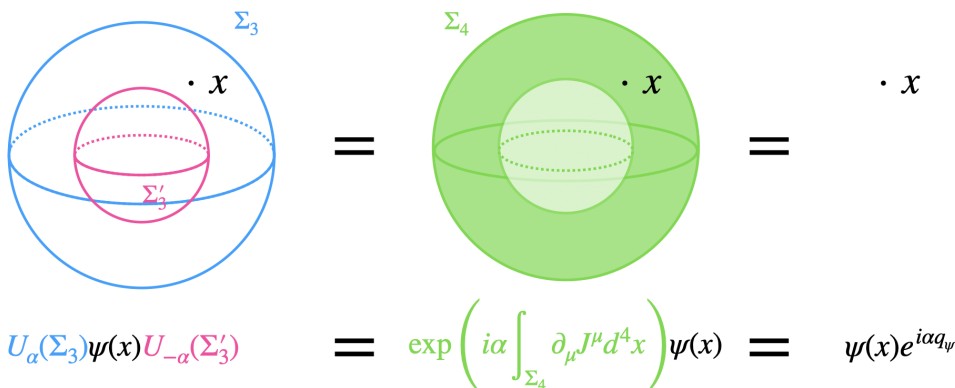

$$U_\alpha(\Sigma_3)\psi(x)U_{-\alpha}(\Sigma_3') \quad = \quad \exp\left(i\alpha\int_{\Sigma_4}\partial_\mu J^\mu d^4x\right)\psi(x) \quad = \quad \psi(x)e^{i\alpha q_\psi}$$

Figure 8: A local operator $\psi(x)$ charged under a $U(1)$ zero-form symmetry and the action of a symmetry defect operator $U_\alpha(\Sigma_3)$ on it by linking as described in the text. One dimension is suppressed.

**Discrete symmetries**   We note also that a useful aspect of this formalism is a unified language for both continuous and discrete symmetries. A discrete $\mathbb{Z}_N$ symmetry doesn't have an associated current because the Noether procedure requires a notion of infinitesimal transformation. However, there are still well-defined SDOs that we can write down and have the expected properties when they act on charged operators,

$$U_{\frac{2\pi k}{N}}(\Sigma_3)\psi(x) = \psi(x)\exp\left(i\frac{2\pi k}{N}q_\psi\mathrm{Link}(\Sigma_3,x)\right). \tag{32}$$

This suffices as a definition in the case of a discrete symmetry by describing how $U(\Sigma_3)$ behaves in arbitrary correlation functions. Of course it may be useful—and depending on the scenario it may be more-or-less easy—to realize the SDO as the integral over $\Sigma_3$ of some local operator. Sometimes we are thinking about a $\mathbb{Z}_N$ subgroup of what is (or began as) a $U(1)$ symmetry, and we can realize $U_\alpha(\Sigma)$ as an integral over a current with the angle restricted to $\mathbb{Z}_N$. This is effectively an operator which measures a global charge (mod $N$), and will be the relevant case for us below with the electric one-form symmetry of electromagnetism.

In other cases when the symmetry is really intrinsically $\mathbb{Z}_N$, it is sometimes useful to introduce an auxiliary $U(1)$-valued field and then project out its dynamics. This becomes an invaluable technique when one wants to understand discrete gauge theories, and we refer to [107] for an expansive discussion of this topic.

## 7.1   One-form symmetries

Yang-Mills theories have long been appreciated to include some gauge-invariant one-dimensional operators known as Wilson loops and 't Hooft loops. These are not local operators because they are defined on a 1-dimensional path $\gamma$ through spacetime which is either a closed loop or an infinite line.[21] Physically a Wilson loop can be seen as the effect of a massive particle of charge $q$ traversing the path $\gamma$, and in the limit where the mass is taken to infinity these Wilson loops capture fully their physical effects. In the Abelian case, the Wilson loop simply integrates the vector potential along this path as

$$W_q(\gamma) \equiv \exp iq\int_\gamma A_\mu dx^\mu. \tag{33}$$

---

[21]Which are closed loops on the one-point compactification of spacetime.

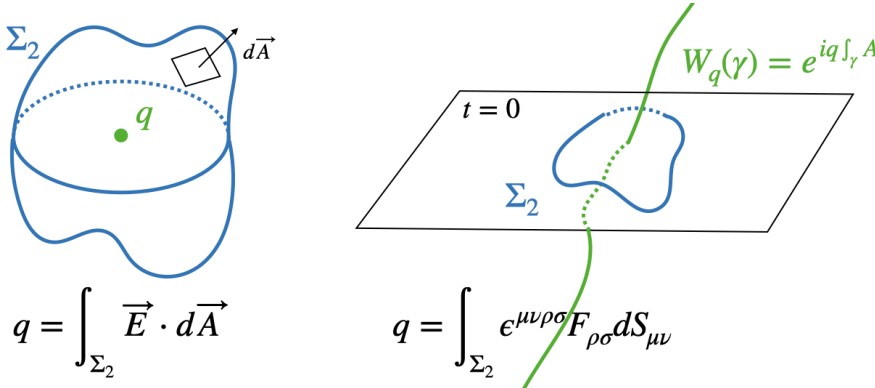

Figure 9: The familiar form of Gauss' law on a timeslice (left) and the more covariant interpretation of the Gaussian surface as a symmetry defect operator $U_\alpha(\Sigma_2)$ acting on Wilson lines charged under a global one-form symmetry.

In the general non-Abelian case the Wilson loops are instead labeled by a representation over which we take the trace $W_R(\gamma) \equiv \operatorname{Tr} \exp i \int_\gamma A_\mu^a T_R^a dx^\mu$. The 't Hooft loops are defined analogously for magnetic representations but with the electromagnetic dual vector potential $A \mapsto \tilde{A}$.[22]

Now the question of which representations our theory allows can be understood field theoretically and gauge-invariantly by examining these line operators and the possible symmetries they might enjoy, which are called one-form symmetries since they act on one-dimensional operators.

We recall Gauss' law in electromagnetism where you think about integrating the electric field over some closed 2-dimensional spatial manifold $\Sigma_2$ and finding some notion of an enclosed charge $Q_{\text{encl}} = \int_{\Sigma_2} \vec{E} \cdot d\vec{A}$. But we can more clearly and more covariantly think about this by recognizing the generalized symmetry structure behind Gauss' law: The Gaussian surface computes a Noether charge for a one-form symmetry!

Pure electromagnetism in fact has both an electric one-form symmetry and a magnetic one-form symmetry. The photon equation of motion and the Bianchi identity reveal the conserved two-index currents which generate these one-form symmetries,

$$\partial_\mu F^{\mu\nu} = 0, \quad \partial_\mu \epsilon^{\mu\nu\rho\sigma} F_{\rho\sigma} = 0. \tag{34}$$

The familiar Gaussian surface can in fact be covariantly upgraded and exponentiated to realize SDOs supported on *any* two-dimensional surface $\Sigma_2$

$$U_\alpha(\Sigma_2) = \exp\left( i\alpha \int_{\Sigma_2} \epsilon^{\mu\nu\rho\sigma} F_{\rho\sigma} dS_{\mu\nu} \right). \tag{35}$$

The SDOs are topological except when they cross Wilson lines and their correlation functions are controlled by

$$U_\alpha(\Sigma_2) W_q(\gamma) = W_q(\gamma) \exp\left( i\alpha q \operatorname{Link}(\Sigma_2, \gamma) \right). \tag{36}$$

This is just the analogue of what we observed above for zero-form symmetries. Now we can talk about the allowed representations in terms of the electric one-form symmetries of the Wilson lines of the theory. Analogously to the argument in terms of gauge transformations, if the

---

[22]For completeness we recall that the dual potential is related to the vector potential in the following nonlocal way. The field strength is $F_{\mu\nu} = \partial_\mu A_\nu - \partial_\nu A_\mu$, and its Hodge dual is $\tilde{F}_{\mu\nu} = \varepsilon_{\mu\nu\rho\sigma} F^{\rho\sigma}$. This dual field strength is related to the dual potential as $\tilde{F}_{\mu\nu} = \partial_\mu \tilde{A}_\nu - \partial_\nu \tilde{A}_\mu$.

electric one-form symmetry is compact ($U(1)$ or a $\mathbb{Z}_N$ subgroup) then there is a transformation by $\alpha = 2\pi$ which should act as the identity

$$U_{2\pi}(\Sigma_2) W_q(\gamma) \equiv W_q(\gamma), \tag{37}$$

and this is seen by the above equation to imply $q \in \mathbb{Z}$, since the linking number is an integer. On the other hand it is conceivable that the electric one-form symmetry is $\mathbb{R}$, though with the same difficulties discussed above that this is thought not to occur in a theory of quantum gravity.

## 7.2 One-form symmetry-breaking

There is an important qualitative difference between 0-form and ($n > 0$)-form symmetries when it comes to their breaking. For a zero-form symmetry, the charged operators are zero-dimensional local operators—precisely the sort which can appear in a Lagrangian density governing the local dynamics of a theory. This means that such symmetries may be explicitly broken by adding a charged operator to the Lagrangian. For a familiar example, if we add a Majorana mass for neutrinos $\mathcal{L} \mathrel{+}= (\tilde{H}L)(\tilde{H}L)/\Lambda$ then we explicitly break the zero-form global $U(1)_L$ lepton number symmetry.

On the other hand, for a higher-form symmetry the charged objects are extended operators. These don't appear in the Lagrangian, and indeed no deformation of the Lagrangian with additional operators can break a higher-form symmetries. Rather, these symmetries can only break if, as you go to high energies, you see that the charged extended operators are realized as dynamical objects in a more-fundamental theory. For example, when you see that (some of) the Wilson lines of electromagnetism are in fact in our universe completed into dynamical charged particles like electrons and protons.

A useful qualitative picture to have of this breaking is of the 'endability' of the Wilson lines [149, 150]. For simplicity we consider an Abelian gauge symmetry where the Wilson lines are labeled by a charge, but the translation to general representations of non-Abelian symmetries is obvious. Consider an 'open' Wilson line

$$W_q(\gamma; x, y) = \exp\left( iq \int_x^y A \right), \tag{38}$$

$$A_\mu \rightarrow A_\mu + \partial_\mu \lambda \quad \Rightarrow \quad W_q(\gamma; x, y) \rightarrow e^{iq\lambda(y)} W_q(\gamma; x, y) e^{-iq\lambda(x)}, \tag{39}$$

which implies that in the infrared the only gauge invariant line operators are closed loops or infinite lines. This is also why it is possible for the SDOs $U(\Sigma_2)$ to have topological correlation functions with the Wilson lines—if $\Sigma_2$ is linked with $\gamma$, it cannot be unlinked by any smooth deformation. Indeed this is the definition of a topological invariant, and this is what breaks when we go to higher energies and see dynamical charged matter.

When we have access to the electron, we can write a gauge-invariant, bilocal line operator

$$\bar{\psi}(y) W_q(\gamma; x, y) \psi(x), \tag{40}$$

which ends on matter fields of charge $q$. Now it is easy to see why the appearance of the dynamical electron breaks the electric one-form symmetries which acted on the integer-charged Wilson lines in the far infrared.

In Figure 10 we depict a time-like Wilson line beginning and ending on a charged fermion, and a Gaussian surface on a time-slice which would measure the charge of the Wilson line. But the surface $\Sigma_2$ can be smoothly deformed up or down the Wilson line and 'slide off' the end, where it can be shrunk to a point. Then the correlation functions of $\Sigma_2$ cannot any longer be topological and depend only on data like $\text{Link}(\Sigma_2, \gamma)$ because this topological linking is no

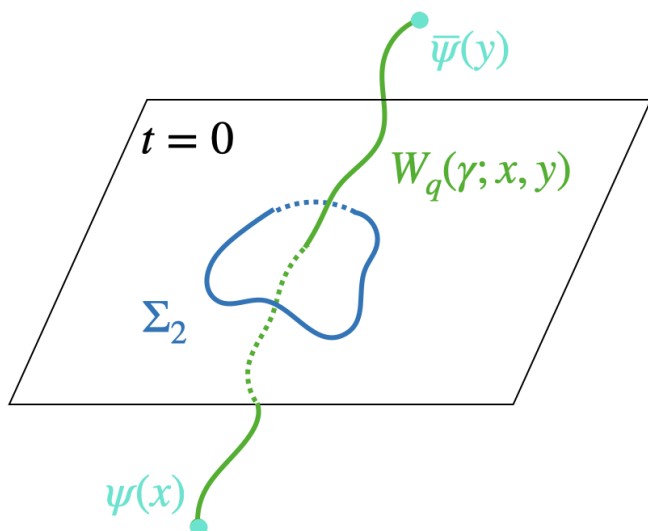

Figure 10: Bilocal line operator one can write cutting a Wilson loop. Such a possibility explicitly breaks any symmetries acting on the Wilson loop because e.g. an SDO on $\Sigma_2$ cannot have non-trivial topological correlation functions any longer when it can smoothly 'slide off' the Wilson line.

longer well-defined. So the appearance of the dynamical $\psi$ field means that any one-form symmetry under which $W_q(\gamma)$ is charged must necessarily be broken. Of course this holds true also for a Wilson line of charge $nq$, $n \in \mathbb{Z}$, which can end on $n$ of these charged fields. But if the charge $q$ of $\psi$ is not the minimum electric charge, there will still be Wilson lines that are not 'endable', and so there may remain an electric one-form symmetry.[23] We now discuss this possibility in more detail, specializing to QED.

### 7.3  Standard Model one-form symmetry

As suggested by the preceding sections, in the full theory of the Standard Model the different global structures correspond to different one-form symmetries. But in fact the latter statement is more general. The existence of a heavy fractionally charged particle implies the existence of an emergent electric one-form symmetry at low energies. We can understand any example universally at low energies where this matches on to an electric one-form symmetry of QED. We reserve a discussion of the electric one-form symmetry in the electroweak phase for Appendix B.

At energies far below the electron mass $E \ll m_e$, none of the Wilson lines of electromagnetism can be 'cut' or 'screened' by dynamical matter, and there is a $U(1)_e^{(1)}$ electric one-form symmetry corresponding to $\theta \in [0, 2\pi)$. This is responsible for Gauss' law.

When we approach energies of order the electron mass $E \gtrsim m_e$, the continuous electric one-form symmetry is necessarily broken. In terms of our Gaussian surface SDOs, the statement is that for general $\theta$, these surfaces will no longer be topological. As shown in [151] we can interpret this violation of the topological invariance of the Gaussian surface as the electric charge being 'screened' and relate it to the running of the fine-structure constant $\alpha$. And indeed we have long appreciated that at these high energies, charges are screened by virtual electron-positron loops. The Uehling potential [152] describing the one-loop photon vacuum polarization tells us the corrected form of the charge $q(r)$ one measures for a Wilson line

---

[23]The case of an $\mathbb{R}$ gauge theory has some slight subtleties in the language one must use to discuss one-form symmetry-breaking, as discussed in Section 6 of [150].

operator of charge $q$ using a Gaussian sphere of radius $r$,

$$q(r \gg m_e) = q\left(1 + e^2 \frac{e^{-2m_e r}}{\sqrt{64\pi^3 m_e r}} + \dots\right), \qquad q(r \ll m_e) = q\left(1 - e^2 \frac{\log m_e r}{6\pi^2} + \dots\right), \quad (41)$$

where we have given the asymptotic forms. Indeed at energies below the electron mass the electric one-form symmetry becomes good exponentially rapidly $dq(r)/dr \approx 0$, while above the electron mass the electric one-form symmetry is clearly broken as the Gaussian surface is far from topological. The question is whether the electron can screen *all* charges, or whether there may remain some unbroken electric one-form symmetry corresponds to fractional charges which the electron cannot screen.

The Gaussian surface in Eq. 35 is normalized such that the electron has $q = 1$, and

$$U_\theta(\Sigma_2) W_\gamma(q) = W_\gamma(q) \exp\left(i\theta q \mathrm{Link}(\Sigma_2, \gamma)\right). \quad (42)$$

Clearly $U_{2\pi}(\Sigma_2)$ acts trivially on the electron, and on every particle with charge a multiple of the electron's. But if there is remaining discrete electric one-form symmetry at energies above the electron's mass, then there are some Wilson lines with $0 < q < 1$ in units of the electron charge. Correspondingly, some $U_\theta(\Sigma_2)$ which act trivially on all Wilson lines of SM representations act non-trivially only on these new Wilson lines, and so remain topological at $E > m_e$. The SM gauge group with the quotient $\mathbb{Z}_n$ has discrete electric one-form symmetry $\mathbb{Z}_{6/n}$ above the electron's mass.

If instead there is no remaining electric one-form symmetry above the electron's mass, as in the case where the SM is embedded in $SU(5)$ in the UV, then every Wilson line has $q \in \mathbb{Z}$. So if we consider $\theta = 2\pi$ then the Gaussian surface will act trivially on *any* operator, and there are no nontrivial $U_\theta(\Sigma_2)$ which remain topological.

So the language of generalized global symmetry conceptually unifies the low-energy experimental signatures by focusing on the symmetry-breaking. In Section 6 above we saw that the SM gauge group could have different global structures. Or it could be that the left-handed quarks $Q_i$ do not actually have the minimum of hypercharge and there is a less-charged particle. Or the hypercharge gauge group could even be $\mathbb{R}_Y$. In any of these cases, the signature in the far infrared where experimentalists work is simply the existence of fractionally charged particles, and we have a unifying statement of what we may learn from such searches as follows

> By discovering a particle with fractional electric charge $q_\psi$ and mass $m_\psi$ we learn the SM has an emergent electric one-form symmetry at $E \ll m_\psi$. If $q_\psi = n/N$ (in units of the electron charge $e$) with $\gcd(n, N) = 1$ then the SM has emergent $\mathbb{Z}_N^{(1)}$ electric one-form symmetry. The unbroken one-form symmetry is measured by the Gaussian surfaces
>
> $$U_k(\Sigma_2) = \exp\left(i2\pi k \int_{\Sigma_2} F\right), \quad (43)$$
>
> with $\theta = 2\pi k$, $k = 1..N$. And in the case where $q_\psi \notin \mathbb{Q}$ then the one-form symmetry is $\mathbb{Z}^{(1)}$, and each $k \in \mathbb{Z}$ makes for a distinct Gaussian surface.

The fact that these Gaussian surfaces remain topological continues to mean that these fractional charges cannot be screened by matter at lower energies. That is, if we surround a heavy fractional charge with a conductor made out of Standard Model particles, it will be unable to prevent a nonzero electric field in its volume.

**Magnetic monopoles**    The low-energy theory of QED also has a magnetic one-form symmetry as seen by the existence of 't Hooft lines and the non-existence of any magnetic monopoles to cut them in the infrared theory. Just as the electric one-form symmetry of the far infrared is always $U(1)^{(1)}$, the magnetic one-form symmetry group is also $U(1)^{(1)}$. But the existence of a discrete electric one-form symmetry above the electron mass controls how the charge of the 't Hooft lines is related to the electron's electric charge. That is, with no electric one-form symmetry, Dirac quantization implies the fundamental magnetic charge is $g = 2\pi/e$. With $\mathbb{Z}_N$ worth of electric one-form symmetry, the quantum of magnetic flux is instead $g = 2\pi N/e$.

## 8  Conclusions

In this work we have called attention to the interesting physics of fractionally-charged particles from both the theoretical and observational perspectives. We have seen that their existence may be tied to the structure of the Standard Model as a quotient group, and correspondingly their discovery would probe nonperturbative aspects of SM physics which could rule out minimal unification schemes from the infrared. More generally, the language of Generalized Global Symmetries provides an interpretation of the existence of heavy, fractionally-charged states in terms of an emergent symmetry possessed by the observed Standard Model.

On the empirical front, we have reinterpreted various LHC searches to derive energy frontier constraints on fractionally-charged particles for a variety of Standard Model representations. In some cases they possess signatures which are well-covered by existing searches (modulo subtleties in particle-detector interactions which we have ignored and deserve further attention), but in other cases the constraints on these exotic, electrically-charged particles from energy frontier searches are weak or nonexistent. Further exploration of possible experimental strategies is clearly warranted to ensure a robust observational program for these striking new particles which could teach us an enormous amount about the universe.

## Acknowledgments

We thank Clay Córdova, Antonio Delgado, and Sungwoo Hong for valuable conversations and Steven Lowette for helpful comments on searches for fractionally charged particles. We are grateful to Sungwoo Hong for comments on a draft of this manuscript.

**Funding information**    The work of SK and AM was supported in part by the National Science Foundation under Grant Number PHY2112540.

## A  Fractionally charged particle partonic cross sections

In this appendix we summarize the partonic cross sections for $\psi_Q$ pair production. The expressions are organized by the spin of $\psi_Q$ and whether or not $\psi_Q$ is charged under $SU(3)$.

We begin with color singlets. For a fermionic $\psi_Q$ with charge $Q_\psi = (\tau_3)_\psi + Y$, where $(\tau_3)_\psi$ is the eigenvalue of the third generator of $SU(2)$ appropriate for $\psi_Q$'s $SU(2)$ representation,

we find:

$$
\frac{d\hat{\sigma}_{EW}(\bar{q}q \to \bar{\psi}_Q \psi_Q)}{d\hat{t}} = \frac{\dim_\psi}{192\,\pi\hat{s}^2}\Bigg( \frac{8e^4 Q_q^2 Q_\psi^2 \left(2M_\psi^4 + 2M_\psi^2(\hat{s} - \hat{t} - \hat{u}) + \hat{t}^2 + \hat{u}^2\right)}{\hat{s}^2}
$$
$$
+ \frac{4g_Z^4 \left(2M_\psi^2\,\hat{s}\,x_L x_R \left(q_L^2 + q_R^2\right) + \left(M_\psi^2 - \hat{t}\right)^2\left(q_L^2 x_R^2 + q_R^2 x_L^2\right) + \left(M_\psi^2 - \hat{u}\right)^2\left(q_L^2 x_L^2 + q_R^2 x_R^2\right)\right)}{\Gamma_Z^2 m_Z^2 + \left(m_Z^2 - \hat{s}\right)^2}
$$
$$
- \frac{8\,e^2 g_Z^2 Q_q Q_\psi \left(m_Z^2 - \hat{s}\right)}{\hat{s}\left(m_Z^4 + m_Z^2\left(\Gamma_Z^2 - 2\hat{s}\right) + \hat{s}^2\right)}\Big(M_\psi^4(q_L + q_R)(x_L + x_R) + M_\psi^2(\hat{s}(q_L + q_R)(x_L + x_R)
$$
$$
- 2\,\hat{t}\,(q_L x_R + q_R x_L) - 2\,\hat{u}\,(q_L x_L + q_R x_R)) + \hat{t}^2\,(q_L x_R + q_R x_L) + \hat{u}^2\,(q_L x_L + q_R x_R)\Big)\Bigg),
$$

(A.1)

$$
\frac{d\hat{\sigma}_{EW}(\bar{q}q' \to \bar{\psi}_Q \psi'_{Q'})}{d\hat{t}} = \frac{\dim_\psi\, e^4(I(I+1) - i_3(i_3 \pm 1))}{192\,\pi\hat{s}^2\sin^4\theta_W}\left(\frac{\hat{t}^2 + \hat{u}^2 + 2M_\psi^2(\hat{s} - \hat{t} - \hat{u}) + 2M_\psi^4}{(m_W^2 - \hat{s})^2 + \Gamma_W^2\,m_W^2}\right). \quad \text{(A.2)}
$$

Here $\hat{s}, \hat{t}, \hat{u}$ are the partonic Mandelstam variables, $g_Z = e/\cos\theta_W$, $q_L, q_R = \tau_3 - Q_q \sin^2\theta_W$ and $x_L, x_R = (\tau_\psi)_3 - Q_\psi \sin^2\theta_W$ factors for $\psi_Q$. The quark factors $Q_q, q_L, q_R$ depend on whether up-type or down-type quarks initiate the collision, while $Q_\psi, x_L, x_R$ depend on which $SU(2)$ representation and hypercharge $\psi_Q$ carries. If $\psi_Q$ is an $SU(2)$ singlet, $x_L, x_R \propto Q_\psi$ so the entire partonic cross section scales as $Q_\psi^2$. Note $\psi_Q$ must have vectorial charge assignment, meaning $x_L = x_R$. The factor of $\dim_\psi$ is the size of $\psi_Q$'s $SU(3)$ representation, should we want to know the electroweak production in that case; $\dim_\psi = 1$ when $\psi_Q$ is a color singlet.

The second expression, $\hat{\sigma}_{EW}(\bar{q}q' \to \bar{\psi}_Q \psi'_{Q'})$, shows the charged current production cross section for $\psi_Q$ in a $SU(2)$ multiplet of size $I(I+1)$. For production via $W^+$, $i_3 = (\tau_\psi)_3$ for the lower charge state within the $\psi$ multiplet and we take the $+$ sign, $i_3(i_3 + 1)$, while for $W^-$ production we take the minus sign and $i_3 = (\tau_\psi)_3$ for the higher charge $\psi$ state.

Keeping the representation the same but switching to scalar $\psi_Q$, the expressions become:

$$
\frac{d\hat{\sigma}_{EW}(\bar{q}q \to \bar{\psi}_Q \psi_Q)}{d\hat{t}} = \frac{\dim_\psi}{192\,\pi\hat{s}^2}\Bigg( \frac{2e^4 Q_q^2 Q_\psi^2 \left(\hat{s}^2 - (\hat{t} - \hat{u})^2 - 4M_\psi^2\hat{s}\right)}{\hat{s}^2}
$$
$$
+ \frac{g_Z^4 x_L^2(q_L^2 + q_R^2)\left(\hat{s}^2 - (\hat{t} - \hat{u})^2 - 4M_\psi^2\hat{s}\right)}{\Gamma_Z^2 m_Z^2 + (m_Z^2 - \hat{s})^2}
$$
$$
- \frac{2g_Z^2 e^2 Q_q Q_\psi x_L^2(q_L + q_R)(m_Z^2 - \hat{s})\left(\hat{s}^2 - (\hat{t} - \hat{u})^2 - 4M_\psi^2\hat{s}\right)}{\hat{s}(\Gamma_Z^2 m_Z^2 + (m_Z^2 - \hat{s})^2}\Bigg),
$$

(A.3)

$$
\frac{d\hat{\sigma}_{EW}(\bar{q}q' \to \bar{\psi}_Q \psi'_{Q'})}{d\hat{t}} = \frac{\dim_\psi\, e^4(I(I+1) - i_3(i_3 \pm 1))}{768\,\pi\hat{s}^2\sin^4\theta_W}\frac{\left(\hat{s}^2 - (\hat{t} - \hat{u})^2 - 4M_\psi^2\hat{s}\right)}{(m_W^2 - \hat{s})^2 + \Gamma_W^2\,m_W^2}, \quad \text{(A.4)}
$$

If $\psi$ carries $SU(3)$ quantum numbers, QCD production $gg \to \bar{\psi}_Q \psi_Q$, $\bar{q}q \to \bar{\psi}_Q \psi_Q$ becomes the dominant mechanism. For fermionic $\psi$ at leading order, we have

$$
\frac{d\hat{\sigma}(gg \to \bar{\psi}_Q \psi_Q)}{d\hat{t}} = \frac{\pi \alpha_s^2 C_2(\psi)}{64 \hat{s}^2} \left\{ -\frac{18\left(2M_\psi^6 - 3M_\psi^4(\hat{t}+\hat{u}) + 6M_\psi^2 \hat{t}\hat{u} - \hat{t}\hat{u}(\hat{t}+\hat{u})\right)}{\hat{s}\left(M_\psi^2 - \hat{t}\right)\left(M_\psi^2 - \hat{u}\right)} \right.
$$

(A.5)

$$
+ \dim_\psi \left( \frac{9C_2(\psi)\left(M_\psi^2 - \hat{t}\right)\left(M_\psi^2 - \hat{u}\right)}{\hat{s}^2} + \frac{2M_\psi^2(3 - 2C_2(\psi))\left(4M_\psi^2 - \hat{s}\right)}{\left(M_\psi^2 - \hat{t}\right)\left(M_\psi^2 - \hat{u}\right)} \right.
$$

$$
\left. \left. - \frac{2C_2(\psi)\left(M_\psi^4 + M_\psi^2(3\hat{t} + \hat{u}) - \hat{t}\hat{u}\right)}{\left(M_\psi^2 - \hat{t}\right)^2} - \frac{2C_2(\psi)\left(M_\psi^4 + M_\psi^2(\hat{t} + 3\hat{u}) - \hat{t}\hat{u}\right)}{\left(M_\psi^2 - \hat{u}\right)^2} \right) \right\},
$$

$$
\frac{d\hat{\sigma}_{QCD}(\bar{q}q \to \bar{\psi}_Q \psi_Q)}{d\hat{t}} = \frac{\pi \alpha_s^2 \dim_\psi C_2(\psi)\left(2M_\psi^4 + 2M_\psi^2(\hat{s} - \hat{t} - \hat{u}) + \hat{t}^2 + \hat{u}^2\right)}{9 \hat{s}^4}.
$$

(A.6)

Here $\dim_\psi$ is the size of the $\psi$ $SU(3)$ representation, $C_2(\psi)$ is the appropriate quadratic Casimir, and we have used $\dim_G C(\psi) = \dim_\psi C_2(\psi)$ to remove all instances of the index $C(\psi)$ and clean up the formulae. For scalar $\psi_Q$, the analogous expressions are:

$$
\frac{d\hat{\sigma}_{QCD}(\bar{q}q \to \bar{\psi}_Q \psi_Q)}{d\hat{t}} = \frac{\pi \alpha_s^2 \dim_\psi C_2(\psi)\left(\hat{s}^2 - (\hat{t} - \hat{u})^2 - 4M_\psi^2 \hat{s}\right)}{36 \hat{s}^4},
$$

(A.7)

$$
\frac{d\hat{\sigma}(gg \to \bar{\psi}_Q \psi_Q)}{d\hat{t}} = \frac{\pi \alpha_s^2 \dim_\psi C_2(\psi)}{128 \hat{s}^2}(\hat{t}^2\hat{u}^2 + M_\psi^4(\hat{t}^2 + \hat{u}^2) - 4M_\psi^6(\hat{t} + \hat{u}) + 5M_\psi^8)
$$

$$
\times \left\{ C_2(\psi)\left( \frac{1}{\hat{s}^2(M_\psi^2 - \hat{t})^2} + \frac{1}{\hat{s}^2(M_\psi^2 - \hat{u})^2} \right) + \frac{2(C_2(\psi) - 1)}{\hat{s}^2(M_\psi^2 - \hat{t})(M_\psi^2 - \hat{u})} \right\}.
$$

(A.8)

# B Electroweak phase one-form symmetry

We have focused on the electric one-form symmetry in the $U(1)_{\text{QED}}$ phase of the SM, but let us turn briefly to the TeV-scale phase, noting that a more technical discussion may be found in [23].

An electric one-form symmetry in the far IR matches on to some electric one-form symmetry of the SM, so the general statement is that there are some Wilson lines which are not endable by the SM matter. The one-form symmetry has rank 1, so we need only one new Wilson line to generate any that is allowed but not realized by the SM matter. We may think of Wilson lines as fusing via the composition of representations.

Then we can always for simplicity choose an $SU(3) \times SU(2)$ singlet representation with some hypercharge. In the cases of the 'global structure' we can think of these as Wilson lines in the representation $R = (1, 1, q)$ with $q = 1, 2, 3$ for $\mathbb{Z}_{6/q}$ electric one-form symmetry. More generally, sticking with this normalization where the left-handed quark doublet has hypercharge $q = 1$, some $q = k/N$ where $\gcd(k, N) = 1$ has $\mathbb{Z}_N$ electric one-form symmetry and $q \notin \mathbb{Q}$ has $\mathbb{Z}$.

By combining these and Wilson lines in the known SM representations one can build the colored or weakly charged representations that give rise to fractionally charged particles as well. However, it is a more subtle task to write down the symmetry defect operators as the

integral of some sort of current, since the centers of $SU(3)_C, SU(2)_L$ are intrinsically discrete. But we know these two-dimensional SDOs measure certain combinations of the non-Abelian center symmetry fluxes and the hypercharge flux. The SM fields do not carry these combinations of charges and so these SDOs act trivially upon them.

In general such operators are known as Gukov-Witten operators [153, 154]. For detailed calculations involved the generalized symmetries it may be useful to introduce auxiliary fields to write the SDOs in a local-looking form, but this goes beyond our remit. For this purpose one would likely wish to begin with the $SU(3)_C \times SU(2)_L \times U(1)_Y$ theory and view the extra $\mathbb{Z}_N$ electric one-form symmetry as deriving from gauging the $\mathbb{Z}_N$ discrete magnetic symmetry of this theory.

The magnetic one-form symmetry of the Standard Model remains group-theoretically $U(1)$ no matter the choice of global structure, but the hypermagnetic monopoles may possess also discrete color- and weak- magnetic fluxes in the case where the global structure is non-trivial. We refer to [19,96] for further detail. Note if we have $\mathbb{R}_Y$ there are no magnetic representations at all, so no magnetic one-form symmetry.

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
