# Peer review of "Fractionally Charged Particles at the Energy Frontier: The SM Gauge Group and One-Form Global Symmetry"

_SciPost Physics, doi:SciPost Phys. 18, 004 (2025)_

## Round 1 · Referee Report · Anonymous (Referee 1) · 2024-8-23

Report

The paper presents a thorough investigation into the phenomenology and implications of fractionally charged particles within SM. If we discover such particles, it would have important implications, offering nonperturbative insights into SM physics and potentially challenging some grand unified theories. The authors emphasize the relevance of searching for these particles, particularly at high-energy colliders like the LHC.

The study carefully examines production mechanisms for these particles at the LHC, considering Dirac fermions and complex scalars in various representations of the SM gauge group. The authors provide detailed analytic expressions for production cross-sections in various cases, highlighting the dependence on the particles' quantum numbers and the energy scale. This analysis is clear and informative.

In discussing collider signatures, the authors reinterpret existing LHC searches, particularly those by CMS and ATLAS, to place bounds on various scenarios. They address the challenges in detecting low-charge particles, suggesting that more targeted experimental strategies may be needed. The interpretation of LHC bounds is reasonably handled, with specific mass limits provided for different representations, though some cases indicate that the current experimental constraints are weaker than expected.

The cosmological implications of discovering fractionally charged particles are also briefly discussed, noting how such a discovery could impact the understanding of relic abundances and the reheating temperature after inflation. This adds context to the study, though it is not the central focus.

The sections on the global structure of gauge theory and generalized global symmetries offer a solid theoretical foundation, exploring how different structures of the SM gauge group could accommodate fractionally charged particles. These discussions are informative and contribute to the overall understanding of the topic. Although the generalized symmetry discussion does not seem to provide new insights, the systematic presentation is sufficiently useful and interesting.

Overall, the paper provides a comprehensive analysis of the production, detection, and implications of fractionally charged particles. I recommend the paper for publication.

Recommendation

Publish (surpasses expectations and criteria for this Journal; among top 10%)

---

## Round 1 · Referee Report · Anonymous (Referee 2) · 2024-9-29

Report

It is well known that there is an ambiguity in the (connected part of) the Standard Model gauge group $G$. The different versions, which differ by a choice of discrete quotient, admit different matter representations (as well as having different 1-form symmetries as the authors review at length), so the potential discovery of BSM particles in exotic representations can rule out certain options, as well as GUTs that match to them. For example, if $G$ is $SU(3)\times SU(2)\times U(1)$ there can be fractionally charged particles, while if $G$ is $SU(3)\times SU(2)\times U(1)/\mathbb{Z}_6$ as predicted by the SU(5) GUT, there cannot.

This paper investigates the phenomenology of such fractionally charged particles at the LHC, and also in cosmology, in part as a means of probing this ambiguity in $G$. A notable phenomenological feature of fractionally charged particles is that, with the exception of a `Higgs portal', the only renormalizable interaction with SM fields is the gauge interaction, which means the leading order collider phenomenology is pair production via the gauge boson (including charged current production in the case of $SU(2)_L$-charged multiplets). The authors systematically study this production in the LHC, together with signatures, for a wide variety of different representations.

Leading order calculations for the signals are implemented analytically, which is appropriate for this kind of exploratory study. Sensitivity to the fractional charge $Q$ is obtained from $dE/dx$ information, which is independent of particle mass for the relevant energies probed; for the case of $SU(2)_L$-charged multiplets, the mass splitting of different states in the multiplet is used to compute decay lengths of the heavier state(s), allowing for further experimental handles from e.g. disappearing tracks in certain cases. These signal models are used to reinterpret a recent CMS search for fractionally charged particles; an ATLAS study of long-lived colored states (in a SUSY context) is also used, as is a further CMS study concerning disappearing tracks which in certain cases can give a stronger bound. The work significantly extends phenomenological studies for fractionally charged particles from the literature, to derive robust leading order estimates of the current constraints on various representations.

The findings are important and clearly presented, making a case for more serious experimental studies and search strategies to look for these states. For states with only (fractional) $U(1)_Y$ charge, the authors find weak bounds - always well below 1 TeV, and for minimal charge of $1/6$ the authors find no meaningful bound at all. For colored states, the bounds are found to be in the $1.5 \div 2$ TeV ballpark, which is consistent with other similar bounds on colored states that do satisfy the conditions in table 2 (e.g. scalar leptoquarks), while the case of $SU(2)_L \times U(1)_Y$-charged color singlet states gives a bound somewhere in between. The authors also include recommendations for how ATLAS and CMS can improve sensitivity to fractionally charged particles going forward, and what improvements are needed in modelling to get better bounds. This is all very constructive for continuing this line of searches at colliders.

For these reasons, I fully recommend this paper for publication, provided the following questions are addressed.

Requested changes

  1. The authors consider complementary constraints coming from the invisible $Z$ decay width, but as far as I can see there is no discussion of other electroweak precision constraints, which are surely also shifted by the fractionally charged particles. For instance, the electroweak gauge boson masses are shifted by 1-loop diagrams, with the fractionally charged state running in the loop; for a state with $U(1)_Y$ charge only for instance, the $Z$ mass would shift but not the $W$ mass. Given an appropriate electroweak input scheme, let's say with $m_Z$ as an input, I would expect the precise $W$ mass measurements at colliders to put some complementary constraints on these particles. Did the authors estimate this effect, and is it relevant given the current precision on $W$ mass measurements? I am especially interested in cases like the $|Q|=1/6$ scalars, for which the $Z$ decay width gives no bound as written in lines 421-2. And how about other EWPOs, such as other Z-pole observables measured precisely at LEP?

Also two very minor comments: 2. In the caption for table 2, the phrase "up to the addition of an integer" appears redundant given the "mod 6" in the column heading (and is indeed confusing, because in these units the equivalence is surely up to the addition of a multiple of 6). 3. Perhaps more of a philosophical question: in line 963-4, it is suggested that `searches for irrationally fractionally charged particles are testing deep principles of UV physics'. I am curious how, given all experimental measurements have finite precision, one could ever test experimentally whether a charge is irrational? We could presumably only ever be a sensitive to a finite number of digits...

Recommendation

Publish (easily meets expectations and criteria for this Journal; among top 50%)

---

## Round 2 · Author Response

We are grateful to the editor and to the referees for their close reading and thoughtful, constructive comments. We have made minor updates to the text to address their questions, as we summarize below.

Indeed, we had thought about electroweak precision constraints and concluded they did not impose any bound but should have included a bit more discussion.

  • For light new species $m_X \ll M_Z$, the limits from anomalous $Z$ decays, such as extra $Z \to$ invisible, provide the strongest constraint. The only benchmark case for which such a small mass is allowed by collider constraints is for $Y=1/6$, and as we had already mentioned such a particle can live in the current uncertainty on $M_Z$.

  • For heavier $m_X$, our setup -- a single fractionally charged particle -- has all of the requirements for an analysis via oblique parameters (specifically, $X$ only couples to SM gauge bosons at tree level). In more modern terms, this setup is a "universal theory" (one where S,T,U are actual observables). See Wells & Zhang 2016 (1510.08462).

  • In our simple setup, many of the oblique parameters are automatically zero: $T$ = 0 (and the same for its derivative $U$), as these measure violations of custodial symmetry and our setup involves no interactions other than gauge interactions. It's $T$ which is sensitive to modifications to the ratio of $M_W, M_Z$, so this means there is no constraint from the $W$ mass measurements. $S = 0$ as well: For scenarios where $X$ only has $U(1)$ charge this is obvious, however it is true even when $X$ has $SU(2)$ interactions. The reason is that $S$ is sensitive to EWSB, given that it involves the two point of a single $SU(2)$ current, while our $X$ is completely blind to EWSB (again stemming from the fact that it has no interactions with $H$).

  • The only oblique parameters which are nonzero are $W$ and $Y$ (which are both insensitive to custodial or $SU(2)$ breaking). To determine the bounding power of these parameters, we can use the results of Cynolter & Lendvai 2008 (0804.4080), rescaling the expressions by a factor of $J(J+1)/(3/4)$ in $W$ (for $X$ in the $J$ $SU(2)$ irrep) or $Y^2_X$ in $Y$ (for $X$ with hypercharge $X$), and compare to the LEP2 bounds they quote. In both cases ($W$ and $Y$) we find there are no meaningful constraints. Specifically, for particles charged only under $U(1)$, the rescaled formulae from Cynolter & Lendvai place bounds well below $M_Z$ and are thus meaningless. For particles with $SU(2)$ charge, $W$ places bounds that are within the realm of validity of an oblique analysis, but the bounds are weaker than the collider constraints, at least for the benchmarks we consider.

  • To derive more accurate bounds for $m_X < m_Z$, we would need to reinterpret LEP2 $e^+e^- \to \bar f f$ data in terms of the full one-loop two point function (rather than just Taylor expanding it as a function of $q^2/m^2_X$). Given that the $Z$ decay constraint already applies in this regime and is superseded by collider bounds in all scenarios with non-zero $SU(2)$ charge, we do not pursue this direction further here.

  • It is worth mentioning that, in more complicated scenarios where there are 2+ fractionally charged particles, if Yukawa interactions between these states and the Higgs is allowed (e.g. if one is an $SU(2)$ doublet, one is a singlet, and their hypercharge differs by $1/2$), then we expect to find $S, T != 0$. Laying out the constraints from EWPO on these slightly-non-minimal but well-motivated models of multiple fractionally charged particles would be an interesting target for future investigation.

---

## Round 2 · List of Changes

• Added a paragraph at the end of Section 3.0 noting that electroweak precision observables do not offer constraints on the benchmark models we consider.
  • Deleted a clause in the caption to Table 2 as suggested by Referee 2.
  • Added a footnote at the end of Section 6.3 on measuring irrational charges vs. rational charges with large denominators.

---

## Editorial Decision

published